# On-surface preparation of coordinated lanthanide-transition-metal clusters

Jing Liu [1,2], Jie Li[1,3], Zhen Xu[1], Xiong Zhou[4], Qiang Xue[1], Tianhao Wu[1], Mingjun Zhong[1], Ruoning Li[1], Rong Sun[4], Ziyong Shen[1], Hao Tang[5], Song Gao [2,4,6], Bingwu Wang[4], Shimin Hou[1,3] & Yongfeng Wang [1,2,6 ✉]

The study of lanthanide (Ln)-transition-metal (TM) heterometallic clusters which play key roles in various high-tech applications is a rapid growing field of research. Despite the achievement of numerous Ln-TM cluster compounds comprising one Ln atom, the synthesis of Ln-TM clusters containing multiple Ln atoms remains challenging. Here, we present the preparation and self-assembly of a series of Au-bridged heterometallic clusters containing multiple cerium (Ce) atoms via on-surface coordination. By employing different pyridine and nitrile ligands, the ordered coordination assemblies of clusters containing 2, 3 and 4 Ce atoms bridged by Au adatoms are achieved on Au(111) and Au(100), as revealed by scanning tunneling microscopy. Density functional theory calculations uncover the indispensable role of the bridging Au adatoms in constructing the multi-Ce-containing clusters by connecting the Ce atoms via unsupported Ce-Au bonds. These findings demonstrate on-surface coordination as an efficient strategy for preparation and organization of the multi-Ln-containing heterometallic clusters.

[1] Key Laboratory for the Physics and Chemistry of Nanodevices and Center for Carbon-based Electronics, Department of Electronics, Peking University, Beijing, China. [2] Division of Quantum State of Matter, Beijing Academy of Quantum Information Sciences, Beijing, China. [3] Peking University Information Technology Institute (Tianjin Binhai), Tianjin, China. [4] Beijing National Laboratory of Molecular Science, College of Chemistry and Molecular Engineering, Peking University, Beijing, China. [5] CEMES, UPR CNRS 8011, Toulouse Cedex 4, France. [6] Institute of Spin Science and Technology, South China University of Technology, Guangzhou, China. ✉email: yongfengwang@pku.edu.cn

anthanide (Ln) series comprises lanthanum (La) and the following 14 *f*-block metal elements from cerium (Ce) to lutetium (Lu). One of the research frontiers devoted to Ln-engaged materials in recent years is the heterometallic compounds containing directly bonded Ln and transition-metal (TM) atoms. Such Ln–TM heterometallic compounds register various potential applications, including high-performance permanent magnets, hydrogen storage materials, luminescent materials, and catalysts, etc.[1–3]. Moreover, the direct (or unsupported) metal–metal bonds between *f*-block and *d*-block elements in the compounds are of great interest in terms of bonding theory. Tremendous effort has been devoted to preparation of coordination compounds involving both Ln metals and TMs by wet chemistry, resulting in a number of Ln–TM heterometallic coordination complexes featuring unsupported *d–f* metal–metal bonds[4–7]. Most of these heterometallic compounds possess metal cores comprising only one Ln atom bonded to one or more TM-containing metalloligand(s), whereas formation of Ln–TM clusters involving multiple Ln atoms as coordination centers in the compounds is scarcely reported[8]. The multi-Ln-containing heterometallic clusters are expected to exhibit novel properties and improved performance owing to the interactions supposed to emerge between the multiple Ln atoms in the clusters such as their magnetic coupling and synergistic effects in catalysis. Moreover, to include multiple Ln atoms in one Ln–TM cluster to build up larger heterometallic aggregates would also narrow the gap between molecular heterometallic complexes and intermetallic solid-state compounds. Therefore, the exploration of the preparation of Ln–TM heterometallic clusters containing multiple Ln atoms is highly desirable.

The formation of the multi-Ln-containing Ln–TM clusters stabilized by unsupported metal–metal bonds requires delicate balance among the various metal–metal and metal–ligand interactions, whose realization by wet chemistry would be rather challenging mainly due to the less controllable reaction conditions in the solvent environment. Fortunately, on-surface coordination chemistry has evolved in the past two decades to be an efficient approach to fabricating a plenty of low-dimensional metal–organic hybrid nanostructures that cannot be achieved by wet chemistry[9–13]. In the on-surface coordination systems, the coordination reactions are confined in the 2D plane due to the absorbate–substrate interactions. Moreover, the ultrahigh vacuum (UHV) condition in which the on-surface constructions of coordination structures are usually carried out ensures a solvent-free reaction environment. Both facts facilitate the controllability of the coordination reactions on surfaces. On-surface coordination thus provides a potential solution to the preparation of the multi-Ln-containing Ln–TM clusters stabilized by unsupported

metal–metal bonds. Recently, a series of Ln–organic hybrids have been constructed via on-surface coordination, in which either single Ln atoms[14–24] or molecule-bridged Ln dimers[17] were identified as the coordination centers. Meanwhile, the achievement of non-lanthanide metallic clusters stabilized by organic ligands on surfaces were also reported[25–30]. Nevertheless, the on-surface preparation of coordination complexes formed by TM-bridged multi-Ln-containing heterometallic clusters has yet to be realized.

In this work, we show the preparation of a series of multi-Ce-containing heterometallic clusters in which the Ce atoms are bridged by Au atoms via the unsupported Ce–Au bonds with an on-surface coordination strategy. By applying various molecular ligands (Fig. 1a), including the pyridines (Py) **1** [1,4-bis(4-pyridyl)-benzene], **3** [1,4-bis(4-pyridyl)-biphenyl], and **4** [1,3-bis(4-pyridyl)-benzene], and the nitrile (CN) **2** [4,4″-dicyano-1,1′:3′,1″-terphenyl], Ce–Au heterometallic clusters coordinated by the organic linkers are obtained on different Au substrates [Au(111) and Au(100)], as revealed by the scanning tunneling microscopy (STM) investigations. The heterometallic clusters containing 2, 3, and 4 Ce atoms in which each pair of the nearest-neighboring Ce atoms are bridged by two Au adatoms assemble into ordered one- or two-dimensional (1D or 2D) structures by forming different coordination motifs with the molecular ligands (Fig. 1b), leading to mono-dispersed domains of the Ce–Au clusters. Density functional theory (DFT) studies reveal the indispensable role of the bridging Au adatoms in stabilizing the multiple Ce atoms within one cluster by forming the unsupported Ce–Au bonds between them. This work demonstrates on-surface coordination as an efficient strategy for preparing and organizing TM-bridged multi-Ln-containing heterometallic clusters stabilized by unsupported *d–f* metal–metal bonds.

## Results
**Coordination structures involving Ce and pyridine ligand 1 formed on Au(111).** Deposition of **1** onto the Au(111) substrate held at room temperature gave rise to a self-assembled structure stabilized by intermolecular hydrogen bonds (Supplementary Fig. 1a). Upon evaporation of Ce atoms onto the **1**-precovered Au(111) substrate held at room temperature, the formation of a single-wall honeycomb network (Fig. 2a) was observed. The hexagonal unit cell of the new structure is highlighted by the white parallelogram in Fig. 2a with a periodicity of 2.6 nm. This 2D network is distinct from that obtained by the coordination of **1** with the Au adatoms on Au(111), that is, the 1D molecular wires in which the molecules are connected by twofold coordination with the Au adatoms (Supplementary Fig. 1b)[31]. Therefore, it is Ce atoms rather than Au adatoms that are engaged in

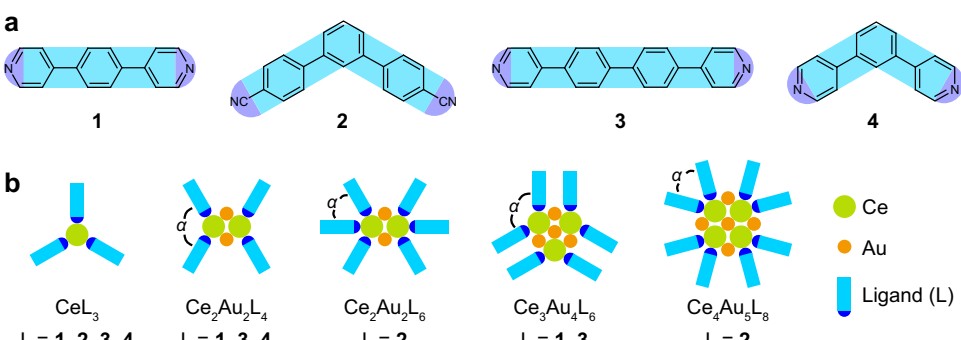

**Fig. 1 Ligands used and coordination motifs achieved in this work. a** Chemical structures of the molecular ligands **1** [1,4-bis(4-pyridyl)-benzene], **2** [4,4″-dicyano-1,1′:3′,1″-terphenyl], **3** [1,4-bis(4-pyridyl)-biphenyl], and **4** [1,3-bis(4-pyridyl)-benzene] employed in this work. **b** Schematic illustration of the coordination motifs achieved in this work. The angles between the ligands approaching to the same Ce atom are marked by α.

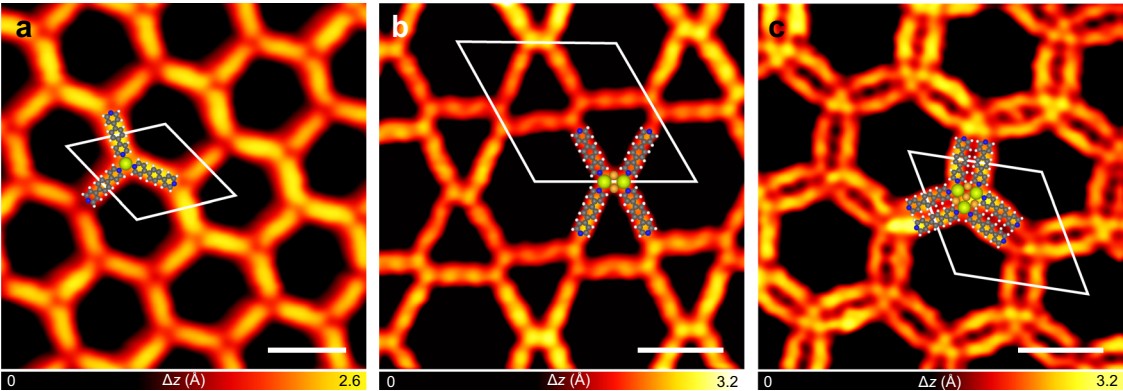

**Fig. 2 Coordination structures involving Ce and 1 formed on Au(111).** STM images of **a** the single-wall honeycomb structure (scanning conditions: bias $V = -10$ mV, tunneling current $I = -30$ pA, imaging temperature $T = 77$ K), **b** the Kagome structure ($V = 5$ mV, $I = 50$ pA, $T = 77$ K), and **c** the double-wall honeycomb structure ($V = 30$ mV, $I = 100$ pA, $T = 77$ K) formed by depositing **1** and Ce on Au(111). The structural motifs constructing the coordination structures are illustrated by the superimposed molecular models. Color code: dark gray for C, white for H, blue for N, light green for Ce, and orange for Au adatom. The unit cells are highlighted by the white parallelograms. Scale bars: 2 nm.

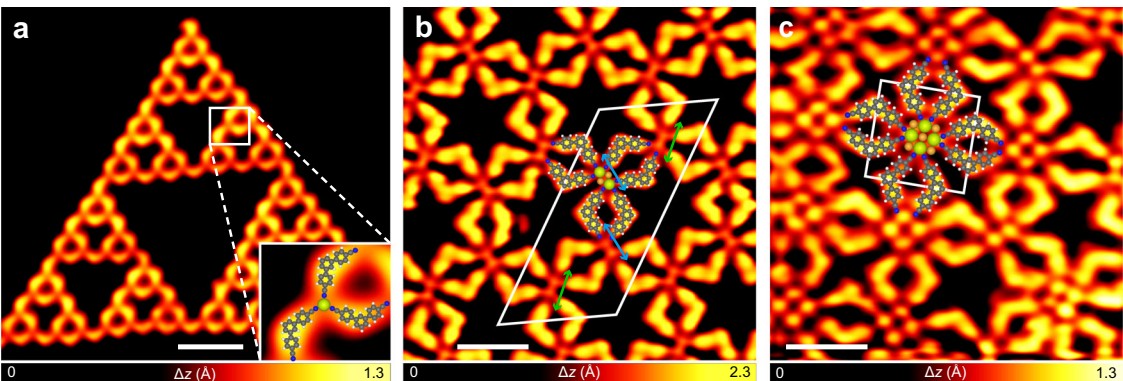

**Fig. 3 Coordination structures involving Ce and 2 formed on Au(111).** STM images of **a** a quasi-fourth-order Sierpiński triangle ($V = 100$ mV, $I = 20$ pA, $T = 77$ K), **b** the three-lobe structure ($V = 10$ mV, $I = 20$ pA, $T = 4.3$ K), and **c** the four-lobe structure ($V = 10$ mV, $I = 200$ pA, $T = 4.3$ K) formed by depositing **2** and Ce on Au(111). Inset of **a**: Magnified image of the Ce(CN)₃ building block of the Sierpiński triangles. The orientations of the Ce dimers in the three-lobe structure are marked by the green and blue arrows in **b**. The unit cells are highlighted by the white parallelograms in **b** and **c**. The structural motifs constructing the coordination structures are illustrated by the superimposed molecular models. Scale bars: **a** 5 nm, **b** 2 nm, and **c** 2 nm.

the single-wall honeycomb structure. Accordingly, a model of the coordination network formed by Ce and **1** is proposed. In this model, three molecules **1** approach to a Ce atom with their pyridine ends to form coordination bonds between the pyridine-N atoms and the Ce atom. This gives rise to the threefold mononuclear coordination node as the building block, denoted as the CePy₃ motif, as illustrated by the superimposed molecular models in Fig. 2a.

Subsequent thermal treatment of the sample at about 355–375 K resulted in the partial desorption of the molecular ligands, which led to a decreased molecule-to-Ce ratio involved in the coordination structures as compared with that in the single-wall honeycomb network, that is, 3:2. As a consequence, two new structures with a lower molecule-to-Ce ratio (1:1) emerged, i.e., the Kagome network (Fig. 2b and Supplementary Fig. 2a) and the double-wall honeycomb network (Fig. 2c and Supplementary Fig. 2b). The STM image of the Kagome structure (Fig. 2b) clearly resolves two identical bright dots (highlighted by the light green dots) at the center of each fourfold node (molecular models superimposed), which are assigned as two Ce atoms. As for the double-wall honeycomb network (Fig. 2c), one can see three bright dots in the STM image (marked by the light green dots) which are identified as a Ce trimer at the center of six molecular ligands (molecular models superimposed). Both the Kagome and

the double-wall honeycomb networks feature the multi-Ce-containing clusters as the coordination centers. The clusters are organized through their coordination with the molecular ligands into the ordered 2D arrays. The 2D networks can be described by the hexagonal unit cells as marked by the white parallelograms in Fig. 2b, c with dimensions of 3.4 and 3.1 nm, respectively.

**Coordination structures involving Ce and nitrile ligand 2 formed on Au(111).** In order to enrich the diversity of the Ce-engaged coordination structures, we employed the nitrile molecule **2** (Fig. 1a) instead of the pyridine molecule **1** as the ligand to construct coordination structures with Ce on Au(111). Without Ce adatoms, molecules **2** were found to assemble into the braid-like structures via intermolecular hydrogen bonds on Au(111) at room temperature (Supplementary Fig. 3). As a comparison, coordinated Sierpiński triangles[11–13] (Fig. 3a and Supplementary Fig. 4a) were obtained by depositing Ce onto the **2**-precovered Au(111) substrate held at room temperature, indicating the engagement of Ce atoms in the fractal structures. The building block of the Sierpiński triangles is the coordination motif formed by one Ce atom approached by three nitrile molecules, denoted as Ce(CN)₃, as illustrated by the superimposed molecular models in Fig. 3a inset.

By depositing more Ce atoms to the sample with the coordinated Sierpiński triangles on the Au(111) substrate followed by thermal treatment of the sample at about 350 K, a three-lobe structure and a four-lobe structure were obtained (Fig. 3b, c and Supplementary Fig. 4b). Figure 3b shows the representative STM image of the three-lobe structure, in which the two bright dots (highlighted by the light green dots) located at the center of six molecules (molecular models superimposed) are assigned as two Ce atoms. The orientation of the Ce dimers, as marked by the green and blue arrows in Fig. 3b, varies regularly within the domain (Supplementary Fig. 5). The unit cell of the three-lobe network is highlighted by the white parallelogram in Fig. 3b with the dimensions of $a = 3.4$ nm, $b = 6.8$ nm, and an angle of about 60° between them. Coexisting with the three-lobe network, the four-lobe coordination structure (Fig. 3c) features a square unit cell as marked by the white frame in Fig. 3c with a periodicity of 2.3 nm. Four bright dots are discernible in the high-resolution STM image (marked by the light green dots in Fig. 3c) which are assigned as four Ce atoms at the center of eight molecules (molecular models superimposed in Fig. 3c).

**Coordination of Ce with different molecules and on different substrates.** To extend the range of conditions applicable for the on-surface preparation of the multi-Ce-containing clusters, as well as to achieve various assemblies of the clusters, the coordination of Ce with different molecules and on varied lattice planes of Au substrate were studied.

Firstly, we employed molecules **3** and **4** (Fig. 1a) as the ligands to construct coordination structures with Ce on Au(111). Both molecules possess pyridine coordination groups but show different backbone structures compared with the pyridine ligand **1**. In addition to the mononuclear coordination structures constructed by the CePy₃ motifs, i.e., the single-wall honeycomb network formed by Ce and **3** (Supplementary Fig. 6a) and the coordinated Sierpiński triangles formed by Ce and **4** (Supplementary Fig. 7a), the multinuclear coordiantion structures were also achieved. Figure 4 shows the three-ordered structures assembled by the coordination between the Ce dimers and ligand **3** or **4**. The coordination of **3** with the two-Ce-containing clusters gives rise to a Kagome network (Fig. 4a and Supplementary Fig. 6b) that shows a larger periodicity (4.5 nm) compared with that formed by **1** (Fig. 2b), and a fishing-net structure with a

hexagonal unit cell of 2.2 nm (Fig. 4b and Supplementary Fig. 6c). The engagement of **4** leads to a chain-like structure with a 1D periodicity of 1.3 nm (Fig. 4c and Supplementary Fig. 7b). Besides these Ce-dimer-based structures, a low yield of the coordination motifs formed by the three-Ce-containing clusters and ligand **3** was also observed (Supplementary Fig. 6d).

Next, the coordination of Ce with either **1** or **2** was investigated on Au(100). As a result, the fishing-net structure (Supplementary Fig. 8a, b) and the double-wall honeycomb network (Supplementary Fig. 8a, c) formed by the pyridine ligand **1** and the two- or three-Ce-containing clusters, as well as the three-lobe (Supplementary Fig. 8d, e) and four-lobe (Supplementary Fig. 8d, f) structures formed by the nitrile ligand **2** and the two- or four-Ce-containing clusters were all obtained on Au(100).

**Structures of the coordinated multi-Ce-containing clusters.** The experimental observation of the coordinated clusters comprising 2, 3, and 4 Ce atoms distinguishes the Ln-engaged on-surface coordination system studied in this work from most of the previously reported ones in which single Ln atoms were identified as the coordination centers[14–24]. To elucidate the structure of the multi-Ce-containing clusters, as well as to figure out the intra-cluster and cluster–ligand interactions, combined experimental and theoretical studies were carried out. The results reveal the heterometallic nature of the clusters, showing that the Ce atoms in the clusters are stabilized by the bridging Au adatoms, as demonstrated below.

First of all, the Ce–Au binary surface system in the absence of molecular ligands is considered. It is documented that the preparation of 2D LnAu₂ intermetallic compounds has been achieved for a number of Ln metals, namely, La[32], Ce[32,33], Gd[34–36], Tb[37,38], Ho[37], and Er[37], by the surface alloying of Ln with the Au(111) substrate. All these LnAu₂ films share the same lattice structure in which each pair of the nearest-neighboring Ln atoms are bridged by two Au atoms[32,34,35,37]. Charge transfer from Ln to Au was found in these systems[34], which is in accordance with the remarkable electronegativity difference between Ln and Au: Ln metals are highly electropositive while Au is the most electronegative metal on the Pauling scale[39]. The electronegativity distinction and the charge transfer are supposed to facilitate the Ln–Au intermixing but suppress the direct Ln–Ln bonding in the binary systems. Similar to Ln elements, other

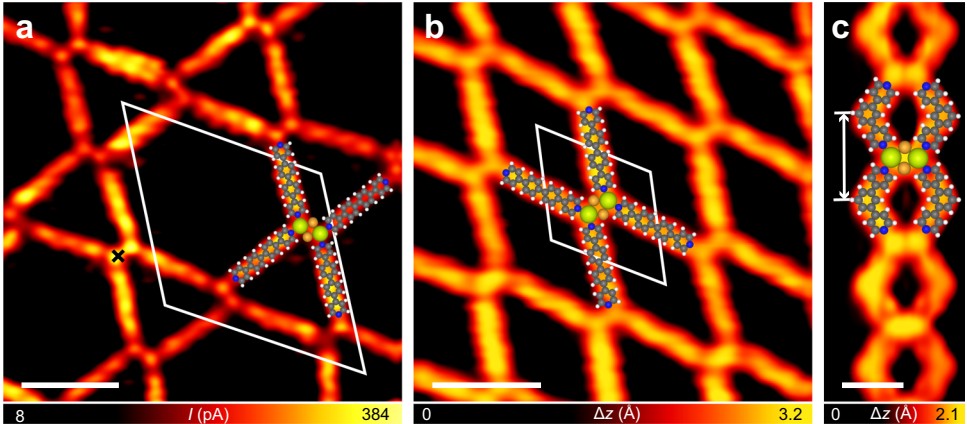

**Fig. 4 Coordination structures involving Ce and 3 or 4 formed on Au(111).** STM images of **a** the Kagome structure (in constant-height mode, $V = 5$ mV, $I = 350$ pA at the point marked by the cross, $T = 77$ K) and **b** the fishing-net structure ($V = 100$ mV, $I = 20$ pA, $T = 77$ K) formed by depositing **3** and Ce on Au(111), and **c** the chain-like structure ($V = -1$ mV, $I = -50$ pA, $T = 4.3$ K) formed by depositing **4** and Ce on Au(111). The unit cells of the Kagome and the fishing-net structures are highlighted by the white parallelograms in **a** and **b**. The 1D periodicity of the chain-like structure is marked by the white arrow in **c**. The structural motifs constructing the coordination structures are illustrated by the superimposed molecular models. Scale bars: **a** 2 nm, **b** 2 nm, and **c** 1 nm.

electropositive metals, e.g., the alkali metals Na[40,41], K[42], and Cs[43,44], have also been reported to form surface alloys with the Au substrates. Therefore, it can be summarized from the literature that the atoms of the electropositive metals, such as Ce, usually intermix with Au atoms rather than directly bond with each other in the binary surface systems.

Next, we take the molecular ligands into account to demonstrate that the Ce atoms coordinated with the molecules are bridged by Au adatoms to form the multinuclear clusters observed in this work. The experimental evidence comes from both the STM and X-ray photoelectron spectroscopy (XPS) results. We conducted STM characterization of the large coordinated clusters with high nuclearities prepared by depositing Ce and ligand **3** on Au(111) followed by 530 K annealing, aiming at comparing the large clusters with the Ce–Au surface intermetallic compound. As shown in Supplementary Fig. 9, clusters comprising seven or more bright dots that are assigned as Ce atoms are connected by the molecular ligands to construct the less ordered coordination structure. Both Ce atoms that are coordinated with the molecules (denoted as $Ce_L$, examples are marked by the blue dots in Supplementary Fig. 9) and those that are surrounded by the other Ce atoms (denoted as $Ce_{Ce}$, examples are marked by the red dots in Supplementary Fig. 9) can be found in the multinuclear clusters. Measurement of the $Ce_L$–$Ce_L$, $Ce_{Ce}$–$Ce_L$, and $Ce_{Ce}$–$Ce_{Ce}$ distances leads to the similar results of $5.0 \pm 0.5$, $5.1 \pm 0.4$, and $5.0 \pm 0.5$ Å, respectively. All these values are comparable to the Ce–Ce separation in the $CeAu_2$ film grown on Au(111), that is, 5.4 Å[32]. Moreover, the arrangement of the Ce atoms in the clusters, although distorted more or less due to the asymmetric interactions of the molecular ligands, roughly resembles the hexagonal lattice of the Ce atoms in the $CeAu_2$ intermetallic compound prepared on Au(111)[32,33]. These results suggest that, on the one hand, the $Ce_{Ce}$ atoms are highly likely to be connected with each other in the same way as that in the $CeAu_2$ surface alloy, i.e., to be bridged by Au adatoms, and on the other hand, the bridging Au adatoms are also responsible for the $Ce_{Ce}$–$Ce_L$ and $Ce_L$–$Ce_L$ connections given the similar Ce–Ce separations and arrangements of the $Ce_{Ce}$ and $Ce_L$ atoms in the large clusters. The analyses provide clues for identifying the two-, three-, and four-$Ce_L$-containing clusters in the ordered coordination structures as the Ce–Au heterometallic clusters.

Furthermore, in order to figure out the chemical state of the Ce atoms in the coordinated clusters, XPS measurements were carried out for the samples with Ce and ligand **2** on Au(111). Different molecule-to-Ce ratios and annealing temperatures were employed for preparing the samples to reproduce the different coordination structures. We found shift of C 1s signal toward higher binding energy by increasing the coverage of Ce (Supplementary Fig. 10a), which is attributed to the charge transfer from the ligands to the Ce atoms caused by the coordination between them. The Ce 3d

spectra exhibit two broad peaks (binding energies at ~885 and 904 eV) that show negligible shifts with the varied molecule-to-Ce ratios and different annealing temperatures of the samples (Supplementary Fig. 10b). The two peaks are characteristic of Ce alloying with Au, as demonstrated by the XPS studies of the Ce–Au surface alloy grown on Au(111) in this work (bottom curve in Supplementary Fig. 10b) and the previous reports[33,45]. The fact that the positions of the Ce 3d peaks are independent from the preparation conditions of the samples suggests the similar chemical states of the Ce atoms in the coordination structures that comprise metallic centers with different nuclearities. This is in agreement with the theoretical findings showing the similar charges carried by the Ce atoms in the different coordination centers (about $+2|e|$ per Ce atom), as described in detail below. It can be concluded from the XPS results that the coordinated clusters are composed of intermixed Ce and Au atoms.

The existence of bridging Au adatoms in the coordinated clusters is also supported by the calculation results. We start from the building block of the three-lobe structure (Fig. 3b), i.e., the coordination motif consisting of a Ce dimer and six nitrile ligands **2**, to interpret the theoretical findings. Firstly, a $Ce_2(CN)_6$ model of the coordination motif in which the coordination center is formed by two directly approached Ce atoms was theoretically tested. The optimized structure of $Ce_2(CN)_6$ in which the simplified nitrile ligands were employed is displayed in Supplementary Fig. 11a. The calculation reveals a positive charge of $+1.98 |e|$ for each Ce atom due to the charge transfer between the Ce atoms and the Au substrate. The approaching of the two positively charged Ce atoms thus leads to a repulsive interaction between them, which is responsible for the positive (i.e., unstable) Ce–Ce binding energy of 0.30 eV, and the electron deficiency between the two Ce atoms as seen in the differential electron density map (Supplementary Fig. 11b). These results exhibit the reduced stability of the structure caused by the direct approaching of the two Ce atoms. Moreover, one can see the slight lift-up of the substrate Au atom between the two Ce atoms (Supplementary Fig. 11a), which is supposed to suppress the lateral movement of the coordinated Ce dimer since such a process requires frequent reconstruction of the substrate. However, in contrast to the calculation results, the tip-manipulation experiment shows the lateral displacements of the coordination motif formed by a Ce dimer and six nitrile ligands as a whole (see Supplementary Fig. 12 and Supplementary Discussion for details), and hence rejects the $Ce_2(CN)_6$ model.

Subsequently, two models with one or two bridging Au adatom (s) located between the two Ce atoms, i.e., $Ce_2Au(CN)_6$ and $Ce_2Au_2(CN)_6$, respectively, were tested. The atomic arrangement of the $Ce_2Au_2$ cluster in the latter model is proposed according to the lattice structure of the $CeAu_2$ surface intermetallic compound grown on Au(111)[32]. The optimized models of the two candidate structures are displayed in Supplementary Fig. 13a and Fig. 5a,

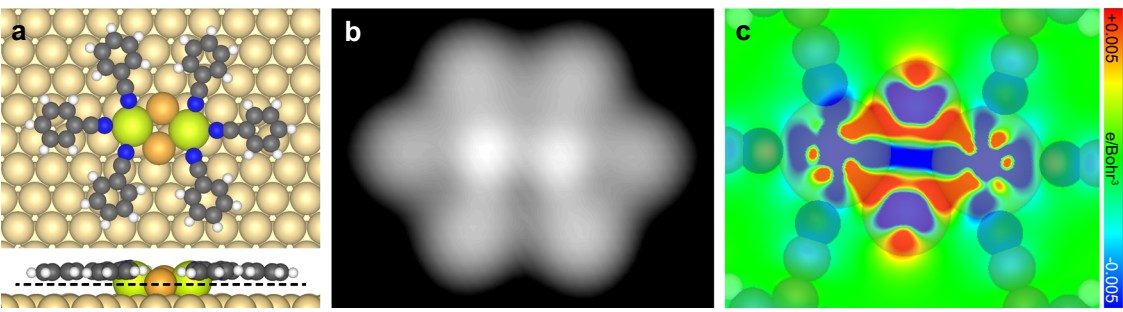

**Fig. 5 Theoretical results of the $Ce_2Au_2(CN)_6$ structure. a** DFT optimized models (upper: top view, lower: side view), **b** simulated STM image, and **c** cross-section of the differential electron density along the black dashed line in **a** of the $Ce_2Au_2(CN)_6$ motif on Au(111). Simplified molecular ligands are employed.

respectively. The calculations reveal that the energy of the $Ce_2Au_2(CN)_6$ [$Ce_2Au(CN)_6$] structure is 1.57 eV (0.75 eV) lower (i.e., more stable) than the total energy of the $Ce_2(CN)_6$ motif plus two (one) non-interacting Au adatom(s). Note that the Au adatom absorbed on the terrace of the substrate is taken as the reference state for the energy calculation here, whereas the Au adatoms on the Au substrate are usually stabilized by re-attaching to the step edges in the real conditions. In the latter context, the energy differences between the proposed structures with and without the bridging Au adatom(s) can be estimated by subtracting the formation energy of the adatom(s) emitted from the step (0.500 eV per Au adatom[46]) from the above-presented results. This gives rise to the energy differences of 0.57 and 0.25 eV for $Ce_2Au_2(CN)_6$ and $Ce_2Au(CN)_6$, respectively, lower than $Ce_2(CN)_6$, which still indicates an energetic preference for formation of the structures with the bridging Au adatom(s). Given that Au adatoms generated from the step and diffusing on the terrace should be well accessible on Au(111) under the reaction conditions[46–50], we propose the most energetically preferred $Ce_2Au_2(CN)_6$ structure among the proposed models as the building block of the three-lobe structure, as illustrated by the superimposed molecular models in Fig. 3b. The similar phenomenon was also found by Yan et al.[30] showing that the evaporated metal atoms were bridged by the intrinsic adatoms to form coordinated heterometallic clusters on surfaces. The Ce–Ce distance in the optimized model of the $Ce_2Au_2(CN)_6$ motif (4.95 Å) is consistent with the experimental value (4.7 ± 0.4 Å) by taking the measurement error into account. Moreover, the simulated STM image of the optimized $Ce_2Au_2(CN)_6$ structure (Fig. 5b) was compared with those of $Ce_2(CN)_6$ (Supplementary Fig. 11c) and $Ce_2Au(CN)_6$ (Supplementary Fig. 13b). As a result, the STM simulation of $Ce_2Au_2(CN)_6$ shows the best agreement with the experimental data, which supports the identification of the $Ce_2Au_2(CN)_6$ structure as the coordination motif to construct the three-lobe network. In both experimental images and STM simulation of $Ce_2Au_2(CN)_6$, the two Ce atoms are resolved as the two bright dots while the Au adatoms are invisible. This can be understood by the larger atomic radius of Ce than Au which leads to a larger height in $z$ direction of the former than the latter when they are adsorbed on the Au substrate, as displayed by the side view of the optimized model (Fig. 5a bottom). The similar situation that only the metal atoms with a bigger size in the heterometallic clusters are discernable by STM was also reported by Yan et al. recently[30].

The higher stability of $Ce_2Au_2(CN)_6$ than $Ce_2(CN)_6$ suggests that the existence of the bridging Au adatoms in the $Ce_2Au_2(CN)_6$ motif facilitates the stabilization of the coordinated cluster. In order to get insights into the stabilization effect of the bridging Au adatoms, Bader charge analysis of the $Ce_2Au_2(CN)_6$ motif was carried out. The calculations uncover positive charges on the Ce atoms (+1.98 $|e|$ per Ce atom, Table 1) and negative

charges on the Au adatoms (−0.47 $|e|$ per Au adatom, Table 1). The whole coordination motif is positively charged (+3.21 $|e|$) and is stabilized by the substrate. These results demonstrate a remarkable charge transfer from the Ce atoms to the Au adatoms and the Au substrate, which is similar to the situation in the Ce–Au binary systems as mentioned above. Therefore, the bridging Au adatoms play an indispensable role in the formation of the two-Ce-containing clusters by serving as the negatively charged "glue" to hold the two positively charged Ce atoms together.

The crucial role of the Au adatoms in stabilizing the $Ce_2Au_2(CN)_6$ motif is further confirmed by the formation of direct Ce–Au bonds in the $Ce_2Au_2$ cluster as demonstrated by our theoretical investigations as follows. The averaged Ce–Au distance ($d_{Ce–Au}$) in the optimized $Ce_2Au_2(CN)_6$ model (Fig. 5a) is 3.02 Å (Table 2), which is smaller than the sum of their covalent radii (3.14 Å)[39], implying the formation of unsupported Ce–Au bonds. The calculation of the total Ce–Au binding energy ($E_{Ce–Au}^{tot}$) in the $Ce_2Au_2(CN)_6$ motif reveals an appreciable energy gain achieved by forming four direct Ce–Au bonds in the multinuclear cluster, that is, 3.03 eV in value (Table 2, also see "Methods" for details of the binding energy calculation). This exhibits the key effect of the Ce–Au interactions in stabilizing the coordination structure. The formation of Ce–Au bond is further supported by the differential electron density map of the $Ce_2Au_2(CN)_6$ structure (Fig. 5c). The cross-section of the differential electron density depicts the electron accumulation (red) and deficiency (blue) due to the formation of the coordination motif (see "Methods" for details of the calculation of differential electron density). It is clearly seen in Fig. 5c that electron density intensifies between the Ce and Au atoms, indicative of the direct Ce–Au bond formation.

As for the cluster–ligand interaction, the optimized model of the $Ce_2Au_2(CN)_6$ motif (Fig. 5a) shows the coordination of three nitrile molecules to each Ce atom in the $Ce_2Au_2$ cluster. The averaged Ce–N distance is 2.53 Å (Table 2), which falls in the range of nitrile–Ce coordination bond length[14,16,22]. The averaged binding energy for each Ce–N bond ($E_{Ce–N}$) is −1.30 eV (Table 2, also see "Methods" for details of the binding energy calculation).

Given the critical role of the bridging Au adatoms in stabilizing the $Ce_2Au_2(CN)_6$ structure, we argue that the coordination centers of all the other multinuclear coordination structures achieved in this work are composed of the Ce–Au heterometallic clusters in which each pair of the nearest-neighboring Ce atoms are bridged by two Au adatoms, as proposed according to the atomic arrangement of the $CeAu_2$ surface alloy prepared on Au(111)[32] (see Supplementary Fig. 14 and Supplementary Discussion for other tested models of the coordinated four-Ce-containing cluster). The optimized models are presented in Fig. 6a–c, showing the $Ce_2Au_2Py_4$ motif as the building block of the Kagome (Figs. 2b and 4a), fishing-net (Fig. 4b), and chain-like (Fig. 4c) structures, while the double-wall honeycomb (Fig. 2c) and four-lobe (Fig. 3c) structures are constructed by the $Ce_3Au_4Py_6$ and $Ce_4Au_5(CN)_8$ motifs, respectively, as illustrated by the molecular models superimposed in Figs. 2b, c, 3c, and 4. The Ce–Ce distances in these models, i.e., 4.52 Å for $Ce_2Au_2Py_4$, 5.56 Å for $Ce_3Au_4Py_6$, and 4.77 Å for $Ce_4Au_5(CN)_8$, are consistent with the experimentally measured values, that is, 4.5 ± 0.3, 5.1 ± 0.5, and 4.9 ± 0.5 Å, respectively. Note that the orientations of the coordination motifs with respect to the substrate as shown in Figs. 5a and 6a–c have all been confirmed by the experimental observations. Bader charges of the Ce and Au atoms in these multinuclear motifs are listed in Table 1, exhibiting that the positively charged Ce atoms are bridged by the negatively charged Au adatoms. The direct Ce–Au interactions in these structures are supported by the $d_{Ce–Au}$ values

**Table 1 Bader charges of the Ce–Au clusters in the coordination motifs.**

| Motif | Charge ($|e|$) | |
|---|---|---|
| | Ce | Au |
| $Ce_2Au_2Py_4$ | +1.88 | −0.59 |
| $Ce_3Au_4Py_6$ | +1.85 | −0.49 (for peripheral Au) |
| | | −0.56 (for central Au) |
| $Ce_2Au_2(CN)_6$ | +1.98 | −0.47 |
| $Ce_4Au_5(CN)_8$ | +1.87 | −0.51 (for peripheral Au) |
| | | −0.76 (for central Au) |

**Table 2 Lengths and Energies of the Ce–Au and Ce–N bonds in the coordination motifs.**

| Motif | $d_{\text{Ce-Au}}$ (Å) | $E^{\text{tot}}_{\text{Ce-Au}}$ (eV) | $d_{\text{Ce-N}}$ (Å) | $E_{\text{Ce-N}}$ (eV) |
|---|---|---|---|---|
| $Ce_2Au_2Py_4$ | 2.98 | −3.38 | 2.59 | −1.33 |
| $Ce_3Au_4Py_6$ | 3.04 (for Ce-peripheral Au) 3.21 (for Ce-central Au) | −5.41 | 2.61 | −1.31 |
| $Ce_2Au_2(CN)_6$ | 3.02 | −3.03 | 2.53 | −1.30 |
| $Ce_4Au_5(CN)_8$ | 2.93 (for Ce-peripheral Au) 3.39 (for Ce-central Au) | −8.69 | 2.55 | −1.17 |

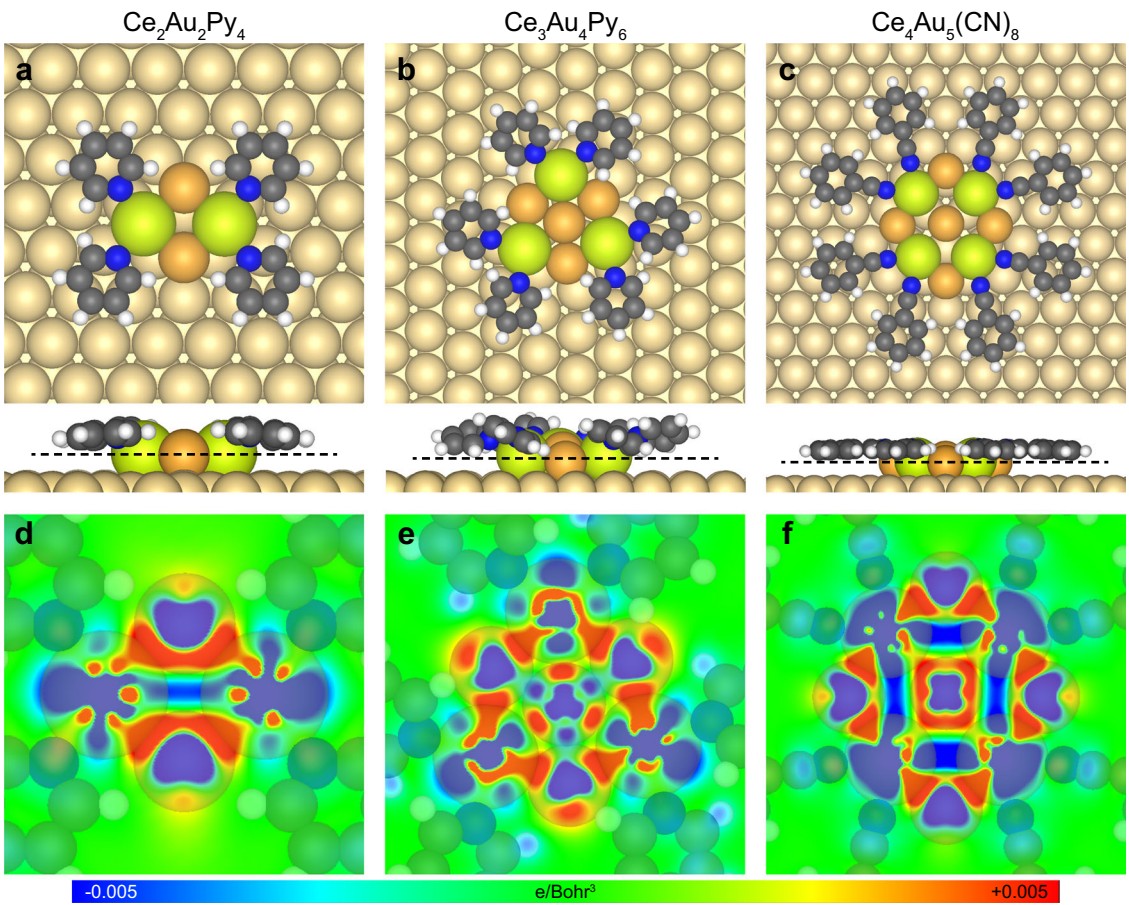

**Fig. 6 Theoretical results of the multinuclear coordination structures.** Optimized models of **a** the $Ce_2Au_2Py_4$, **b** the $Ce_3Au_4Py_6$, and **c** the $Ce_4Au_5(CN)_8$ motifs (upper: top view, lower: side view). Cross-sections of the differential electron densities of **d** the $Ce_2Au_2Py_4$, **e** the $Ce_3Au_4Py_6$, and **f** the $Ce_4Au_5(CN)_8$ motifs along the black dashed lines in **a**, **b** and **c,** respectively.

(Table 2) that are comparable with the sum of covalent radii of Ce and Au (3.14 Å)[39], as well as the large total Ce–Au binding energies ($E^{\text{tot}}_{\text{Ce-Au}}$ in Table 2, see "Methods" for details of the binding energy calculations). The differential electron density maps of the three multinuclear motifs (Fig. 6d–f) provide another evidence for the unsupported Ce–Au bonds by presenting the electron density accumulation between the Ce and Au atoms in these structures. As for the cluster–ligand interactions in the three multinuclear motifs, it is seen in the optimized models (Fig. 6a–c) that the coordination interactions are formed between the molecular ligands and the Ce atoms in the Ce–Au clusters. The calculated Ce–ligand distances and binding energies are summarized in Table 2.

## Discussion

Firstly, we compared the multinuclear coordination motifs formed by the pyridine ligands **1**, **3**, and **4**, i.e., $Ce_2Au_2Py_4$ and $Ce_3Au_4Py_6$, with those constructed by the nitrile ligand **2**, i.e.,

$Ce_2Au_2(CN)_6$ and $Ce_4Au_5(CN)_8$. It is found that the nitrile-engaged multinuclear motifs show smaller angles between the ligands approaching to the same Ce atom [marked by $\alpha$ in Fig. 1b; $\alpha \sim 60$–70° for $Ce_2Au_2(CN)_6$ and $Ce_4Au_5(CN)_8$] than the ones formed by the pyridine ligands ($\alpha \sim 120°$). This can be explained by the smaller volume of the nitrile group than the pyridine group, which allows for the approaching of the nitrile ligands to the same metal atom with a relatively small coordination angle without introducing significant steric hindrance. The different coordination behaviors of the pyridine and nitrile ligands with the Ce–Au clusters give rise to the various ordered coordination assemblies observed in this work, showing the feasibility of utilizing different molecular ligands to tune the organization and periodicity of the Ce–Au cluster arrays.

Next, we discuss the possible mechanism for the formation of the multinuclear coordination structures. All the multinuclear coordination structures achieved in this work feature relatively low molecule-to-Ln ratios (no more than 3:2). Each Ce atom in these structures coordinates with two or three molecular ligands.

As a comparison, relatively high molecule-to-Ln (or molecular coordination site-to-Ln in the case of chelating ligands being engaged) ratios (3:2 or higher) were frequently found for the previously reported Ln-engaged mononuclear coordination structures constructed on surfaces[14–24], in which the single Ln atoms afforded multi (≥3)-fold coordination with the molecular ligands. Accordingly, we propose that the low molecule-to-Ln ratios employed for the preparation of the metal-organic hybrids in this work should play a key role in the formation of the multinuclear coordination structures.

One may expect two types of coordination products in the samples with a molecule-to-Ln ratio ≤3:2, i.e., the mononuclear structures with low coordination numbers (namely, two- or threefold coordination), and the multinuclear structures. The comparison of the stability of these two types of coordination structures can be carried out based on the calculation results shown above. For the two- and threefold mononuclear structures, the stabilizing energy per Ce atom due to the formation of the coordination bonds with the ligands can be estimated as about −2.6 and −3.9 eV, respectively, by taking the averaged Ce–N binding energy of the coordination structures concerned in this work for evaluation. As for the multinuclear structures constructed by the $Ce_2Au_2Py_4$, $Ce_3Au_4Py_6$, $Ce_2Au_2(CN)_6$, or $Ce_4Au_5(CN)_8$ motifs, the stabilizing energy per Ce atom contributed by the bonding with both the molecular ligands and the Au adatoms ranges from −4.4 to −5.4 eV. As a conclusion, the multinuclear coordination structures are thermodynamically preferred at low molecule-to-Ln ratios. Thus, the reduction in the molecule-to-Ln ratio of the sample, which can be achieved by either deposition of additional Ce or molecular desorption caused by thermal treatment of the sample, can serve as a driving force for the emergence of the multinuclear coordination structures.

Moreover, thermal treatment of the sample is also necessary for the preparation of the multinuclear structures. In addition to tuning the molecule-to-Ln ratio by the thermal-induced molecular desorption as mentioned above, the thermal treatment also provides energies for the diffusion and rearrangement of the molecules and metal adatoms on the surface, which is indispensable for obtaining the thermodynamically preferred multinuclear products. Otherwise, coordination structures with lower stability may emerge, such as the threefold mononuclear structures (namely, the single-wall honeycomb network formed by **1** and the Sierpiński triangles formed by **2**) obtained at room temperature in this work.

Last but not least, we have taken the three- and four-lobe structures constructed by the $Ce_2Au_2(CN)_6$ and $Ce_4Au_5(CN)_8$ motifs, respectively, as examples to explore the magnetic and electronic properties of the Ce–Au clusters by STS experiments. However, we failed to get any magnetic signals of the Ce–Au clusters by high-resolution d$I$/d$V$ measurement. The similar situations have been reported for some surface-confined double-decker lanthanide phthalocyanine molecules whose $4f$ electron states were elusive to be detected by STS measurement[51–54]. The reason may lie in the inner-core nature of the $4f$ orbitals which leads to the little contribution of the $4f$ electron of Ce to the tunneling current. We neither found any Ce-specific d$I$/d$V$ feature in the large range (−4 to 3 V) d$I$/d$V$ spectra. It may be explained by the strong hybridization between the Au substrate and the Ce–Au clusters.

To summarize, a series of heterometallic clusters containing 2, 3, and 4 Ce atoms bridged by Au adatoms were prepared with an on-surface coordination strategy in UHV. STM investigations reveal the formation of ordered coordination structures based on the well-defined Ce–Au clusters and the pyridine (**1**, **3** and **4**) or

nitrile (**2**) ligands on the Au(111) and Au(100) substrates, yielding orderly arranged mono-dispersed Ce–Au cluster arrays. The Au adatoms serve as the atomic "glue" to bridge the Ce atoms via the unsupported Ce–Au bonds, as demonstrated by the DFT calculations, which play a crucial role in stabilizing the heterometallic clusters. These results shed light on the application of on-surface coordination as an efficient strategy for the preparation and organization of TM-bridged heterometallic clusters containing multiple Ln atoms which are of great significance at both fundamental and applied levels.

## Methods

**Sample preparation, STM, and XPS measurements.** The samples were prepared in UHV preparation chambers connected with either the STM or XPS chamber. Au (111) and Au(100) surfaces were cleaned by repeated cycles of Ar$^+$ sputtering and annealing. Molecules **1**, **2**, **3**, and **4** were thermally sublimated from the tantalum boats. Ce was evaporated from an electron-beam evaporator at about 1420 K. The thermal treatment of the samples was carried out with a temperature control within ±20 K. All STM experiments were carried out with a Unisoku UHV-STM with a base pressure lower than $3 \times 10^{-10}$ Torr in the system. STM images were acquired in constant-current mode if not otherwise specifically stated and were processed by the WSxM software[55]. All XPS experiments were carried out in a SPECS system with a monochromatic Al $K\alpha$ X-ray source ($h\nu = 1486.6$ eV). The base pressure of the system is lower than $5 \times 10^{-10}$ Torr. The Au $4f_{7/2}$ peak of a clean Au(111) substrate was used to calibrate the spectrometer before experiments. The Au $4f_{5/2}$ peak of the Au(111) substrate at 87.7 eV was employed as an internal standard to calibrate the binding energy scale for this work.

**Calculation methods.** All DFT calculations were performed using the projected augmented-wave (PAW) pseudopotentials[56,57], as implemented in Vienna Ab initio Simulation Package (VASP)[58]. The exchange-correlation energy was calculated with the opt-B88 functional in order to take the van der Waals (vdW) dispersive correction into consideration[59–61]. Scalar relativistic effect which includes the mass-velocity and Darwin correction terms, as implemented in VASP[62], was taken into account. The potential of Ce was generated with 12 valence electrons including the $4f$ electron. The cut-off energy was 400 eV for the Kohn–Sham wave functions. A vacuum space of 10 Å was applied between neighboring structures to ensure their decoupling. The initial geometries of the Ce–Au clusters in the models for optimization were set either according to the lattice structure of the CeAu$_2$ intermetallic compound prepared on Au(111)[32], or by referring to the parameters achieved in other optimized structures studied in this work. The geometry was fully optimized when the maximum forces fell down below 0.02 eV Å$^{-1}$. A Γ centered grid was applied for the Brillouin zone sampling.

The total Ce–Au binding energy ($E_{Ce-Au}^{tot}$) of the Ce$_m$Au$_n$L$_k$ (L for ligand) motif is calculated as below

$$E_{Ce-Au}^{tot} = E_{Ce_mAu_nL_k/sub} - E_{Ce_mL_k/sub} - E_{Au_n/sub} + E_{sub} \qquad (1)$$

in which $E_{Ce_mAu_nL_k/sub}$, $E_{Ce_mL_k/sub}$, $E_{Au_n/sub}$, and $E_{sub}$ refer to the energies of the Ce$_m$Au$_n$L$_k$ motif on the substrate, the Ce$_m$L$_k$ fragment on the substrate, the Au adatoms on the substrate, and the Au(111) substrate, respectively. Here, the Au adatom on the terrace of the substrate is taken as the reference state for the energy calculation. The total Ce–Au binding energy relative to the Au atom at the step edge can be estimated by adding the formation energy of the Au adatoms emitted from the step on Au(111) (0.500 eV per atom[46]) to $E_{Ce-Au}^{tot}$. The so-achieved results still represent considerable stabilization effect of the Ce–Au bonds.

The Ce–N binding energy ($E_{Ce-N}$) of the Ce$_m$Au$_n$L$_k$ motif is calculated as below

$$E_{Ce-N} = (E_{Ce_mAu_nL_k/sub} - E_{Ce_mAu_n/sub} - E_{L_k/sub} + E_{sub})/k \qquad (2)$$

in which $E_{Ce_mAu_n/sub}$ and $E_{L_k/sub}$ refer to the energies of the Ce$_m$Au$_n$ cluster on the substrate and the molecular ligands on the substrate, respectively.

The differential electron density (d$\rho$) is calculated as below

$$d\rho = \rho_{tot} - \rho_{Ce} - \rho_{Au} - \rho_L - \rho_{sub} \qquad (3)$$

where $\rho_{tot}$, $\rho_{Ce}$, $\rho_{Au}$, $\rho_L$, and $\rho_{sub}$ correspond to the electron densities of the total system, the Ce atom, the Au adatom, the molecular ligand, and the substrate, respectively.

## Data availability

The data that support the findings of this study are available within the paper and its supplementary information file. Raw data are available from the corresponding author upon reasonable request.

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

## Acknowledgements

This work was supported by the Ministry of Science and Technology (2018YFA0306003, 2017YFA0205003) and National Natural Science Foundation of China (22001017, 21972002, and 21991132). DFT calculations were carried out on TianHe-1A at National Supercomputer Center in Tianjin and supported by high-performance computing platform of Peking University.

## Author contributions

Y.W. designed the experiment; J. Liu, Z.X., Q.X., and R.L. conducted the STM experiments; J. Li., R.S., H.T., and B.W. carried out DFT calculations; X.Z,. T.W., and M.Z. conducted the XPS experiments; J. Liu, Q.X., S.G., Z.S., B.W., S.H., and Y. W. analyzed the data; S.G., Z.S., B.W., and S.H. participated in manuscript writing; J. Liu, Q.X., and Y.W. wrote the manuscript. All authors discussed the results and commented on the manuscript.

## Competing interests

The authors declare no competing interests.
