## [Peer Review File · Nature Communications]

REVIEWER COMMENTS

Reviewer #1 (Remarks to the Author):

Liu et al. present an investigation of Ce-clusters with organic ligands on Au(100) and Au(111) via on-surface synthesis. Analysis is based on scanning tunneling microscopy (STM) experiments and density functional theory (DFT) computations. The core finding of the study is the isolation of clusters containing more than two Ce atoms on the surface – a novum.

The motivation for the work is well outlined in the introduction although the possible application of such structures could be spelled out more clearly (line 35). Nevertheless, as a surface science study exploring new avenues, the prospect of deriving these interesting, multimetallic structures is motivation enough.

On-surface synthesis is still a striving field and the literature on multi-metal core structures is rather scarce, thus this contribution could present a significant next step.

The structures are all well analyzed based on STM measurements and the images support the presented structural interpretations which seem reasonable at first glance. Nevertheless, since STM does not indicate the type of element hidden underneath the features measured, the structural proof is indirect.

The contribution from DFT is the proposal that the Ce-organic on-surface complexes are held together by Au adatoms. And here, the evidence is not strong enough. STM does not give an indication of Au adatoms (the authors argue that they cannot be seen and give a reference for this statement). Thus, the evidence for Au-coordination is purely from theory. And the evidence from DFT is essentially one total energy calculation stating that the proposed Ce(n)Au(m)-complexes are more stable than Au adatoms on the surface. This is an indication, but not a strong proof. It could well be that the calculated complexes are less stable than complex plus surface and the presence of adatoms is an ad-hoc assumption. Furthermore, no discussion about a possible mechanism for the formation of this unusual complex is given.

The computational method section also raises several questions. The authors claim to have used PBE as density functional and treat dispersion interaction by optB88. This does not make sense since the latter is a vdW-density functional which is used standalone and is not combined with PBE. Thus, it is unclear how the numbers in the manuscript are derived. It also does not lend confidence in the computational section altogether.

The question if the 4f electrons of Ce are really not participating in the bonding (as implicitly assumed by using a large core pseudopotential) also needs investigation.

Plotting the difference density w.r.t. the atomic densities leads to a picture where many aspects are mixed. Thus, it is not clear at all where the density change comes from: bonding or polarization. The pictures should be plotted w.r.t. the free molecule and the free surface which is the standard procedure in the field (and even then it is often not clear what one sees).

What has not been discussed at all is how the optimized structures were derived. Have several reasonable starting structures been assumed? How was the Ce-Ce distance chosen on the surface (which seems very large in Figure S6)? Also, the convergence criterion is not strict enough (<0.02 eV/Å).

I see the need for direct experimental evidence for the proposed structures by revealing the nature of the atoms in the on-surface structure. This could for example be derived by XPS or NEXAFS investigations. Up to now, the structural determination relies too much on interpretation of (partially weak) indirect evidence.

Reviewer #2 (Remarks to the Author):

Wang and coworkers report an elegant approach for the controlled on-surface formation of ligand-stabilized heterometallic Ce/Au clusters with two, three and four Ce atoms. The different cluster sizes are realized by variation of the ligand structure (nitrile and pyridine based) and the

preparation procedures, resulting in different ligand-to-metal ratios. The diversity of the structures and the level of control are unprecedented and very impressive. The high-quality STM data clearly show the positions of the Ce atoms and the ligands. The presence of additional Au atoms in the clusters is derived from plausibility considerations and is supported by DFT calculations. The DFT calculations also reveal that the Ce-Au bond has partial ionic character, owing to the large electronegativity difference. This work is highly original and carefully composed. Publication in Nature Communications is recommended after some minor issues have been addressed.

1. On page 10, it is stated that the Ce₂Au₂L₆ structure (L = ligand) is lower in energy than the Ce₂L₆ structure plus two non-interacting Au adatoms by 1.94 eV. From this, it is concluded that the Ce₂Au₂L₆ structure is energetically stabilized. However, to make this conclusion, the non-interacting Au adatoms are not the correct point of reference: If the Au adatoms gain even more energy by re-attaching to a step edge (where they typically originate from), then the Ce₂Au₂L₆ structure would not be stabilized. This can easily be understood by considering that the equilibrium concentration of adatoms decreases when the temperature is lowered. When the adatoms return to the step edges, also the Ce₂Au₂L₆ structure would break apart if the Au adatoms gain more energy by binding to the step edge. (Unless Ce₂Au₂L₆ is a kinetically trapped state, but this is not what the authors mean.) Therefore, the energy gain must be calculated relative to the Ce₂L₆ structure plus two Au atoms at a step edge. A typical literature value for the energy of formation of an Au adatom on Au(111) is 0.61 eV. With this value, the energy gain by formation of the Ce₂Au₂L₆ structure would be reduced to 0.72 eV, which still represents a considerable stabilization.

2. On pages 12 and 13, the total calculated Ce-Au binding energies in Ce₂Au₂L₆ and the other structures are presented. According to the methods section (page 20), Au adatoms are used in the reference state. While the resulting values are certainly meaningful, they do not really describe the energetic stabilization of the clusters, as explained in the previous comment. To confirm the energetic stabilization of the structures, referencing to Au atoms at step edges is again more reasonable. This would increase all energy values by around 1.2 eV, but all of them would still be negative. Since the authors emphasize that the Ce-Au interactions stabilize the clusters (rather than Au-Au or Ce-Ce interactions), it may also be instructive to compare with the energies of the corresponding homometallic clusters (Au₄ and Ce₄).

3. Methods: Since Au and also Ce show substantial influence of relativistic effects, it should be mentioned whether a scalar relativistic (SR) or fully relativistic (FR, spin-orbit coupling included) approach was used. There can be substantial energy differences between SR and FR results (see e.g. P. Pyykkö, Annu. Rev. Phys. Chem. 63 (2012) 45-64).

4. On page 18, it is stated that high molecule-to-Ln ratios (3:2 or higher) were frequently found in previous mononuclear coordination structures on surfaces. In this context, it is interesting to note that a 1:1 molecule-to-Ln ratio was achieved by reducing the number of molecules (by a factor of 5) through an on-surface oligomerization of dinitriles, resulting in a lanthanide superphthalocyanine.

5. Temperatures for the thermal treatment are reported as approximate values ("about 350 K"). Since precise temperatures may be important for the reproducibility of the experiments, error margins should be reported.

6. It should be mentioned whether the differential electron density map in Figure 4c and 5d-f shows a projection or a cross section.

J. Michael Gottfried

Reviewer #3 (Remarks to the Author):

Report for "On-Surface Preparation of Coordinated Lanthanide-Transition-Metal Clusters"

The paper describes a series of heterometallic clusters containing 2, 3 and 4 Cerium atoms bridged by Au adatoms prepared under UHV. The authors describe the formation of coordination structures based on Ce-Au clusters and pyridine or nitrile linkers on Au(111) and Au(100). Interestingly, Au adatoms serve as atomic "glue" to bridge the Ce atoms, as demonstrated by the DFT calculations, which play a crucial role in stabilizing the heterometallic clusters.

The structures are analyzed by high-resolution STM, complemented by DFT calculations. The

quality of the data is very good, though before I recommend publication some points have to be considered by the authors, as similar works have been recently reported (see for instance *Angew. Chem. Int. Ed.* 57, 4617-4621 (2018)) where Bi₃Cu₄ and Bi₇Cu₁₂ nanoclusters using pyridil linkers in 2D metal-organic networks have been observed.

- My main concern is related to the lack of information regarding the electronic properties of these systems (or at least of some of them), which I think are very interesting, as Ln-TM clusters involving multiple Ln atoms as coordination centers are scarcely reported, and needed to publish this work in a high-level impact factor journal such as *Nature Communications*.

- I am aware of the difficulties of experimentally proving the presence of Au adatoms as "glue" between Ce atoms. Is there any difference between the simulated STM image shown in Figure 4b (where Au adatoms are having into account in the calculation) and the simulated STM image where those Au are not present? If yes, the authors should comment about it.

- Experimental Ce-Ce distances in the clusters should be included and compared to the theoretical values shown in the manuscript.

Dear Reviewers,

Thank you for your critical advice and comments to our manuscript entitled “On-Surface Preparation of Coordinated Lanthanide-Transition-Metal Clusters”, submitted for publication as an article in *Nature Communications*. Attached please find the revised manuscript.

In the following, we’ll address one by one the comments or questions. All the responses are typed in blue. All major changes in the text are highlighted in red so that you can feasibly identify where we have made the corrections and modifications.

Thank you for your reconsideration.

Sincerely,

Yongfeng Wang
Department of Electronics
Peking University

Reviewer #1

Comment:

Liu et al. present an investigation of Ce-clusters with organic ligands on Au(100) and Au(111) via on-surface synthesis. Analysis is based on scanning tunneling microscopy (STM) experiments and density functional theory (DFT) computations. The core finding of the study is the isolation of clusters containing more than two Ce atoms on the surface – a novum.

The motivation for the work is well outlined in the introduction although the possible application of such structures could be spelled out more clearly (line 35). Nevertheless, as a surface science study exploring new avenues, the prospect of deriving these interesting, multimetallic structures is motivation enough.

Author reply:

We have revised the sentence mentioned by the reviewer to put more emphasis on the potential applications of the Ln-TM complexes as follow:

Paragraph 1, Page 3: “Such Ln-TM heterometallic compounds register various potential applications including high-performance permanent magnets, hydrogen storage materials, **luminescent materials and catalysts**, etc¹⁻³.”

Comment:

On-surface synthesis is still a striving field and the literature on multi-metal core structures is rather scarce, thus this contribution could present a significant next step.

The structures are all well analyzed based on STM measurements and the images support the presented structural interpretations which seem reasonable at first glance. Nevertheless, since STM does not indicate the type of element hidden underneath the features measured, the structural proof is indirect.

The contribution from DFT is the proposal that the Ce-organic on-surface complexes are held together by Au adatoms. And here, the evidence is not strong enough. STM does not give an indication of Au adatoms (the authors argue that they cannot be seen and give a reference for this statement). Thus, the evidence for Au-coordination is purely from theory. And the evidence from DFT is essentially one total energy calculation stating that the proposed Ce(n)Au(m)-complexes are more stable than Au adatoms on the surface. This is an indication, but not a strong proof. It could well be that the calculated complexes are less stable than complex plus surface and the presence of adatoms is an ad-hoc assumption. ...

Author reply:

Fig. R1. Chemical structures of the molecular ligands (a) **1**, (b) **2**, (c) **3** and (d) **4** used in this work.

We thank the reviewer for the helpful comment. In order to make the conclusion of the formation of the coordinated Ce-Au heterometallic clusters in this work more solid, we have made major revisions to the manuscript by (1) adding discussion based on literature research, (2) presenting results of the additional XPS and STM experiments, and (3) improving theoretical calculations. The revisions are summarized as follows:

1. Discussion based on literature research

We have added the discussion about the alloying of the electropositive Ln metals with the electronegative Au substrates based on literature research. This is to show that the Ce atoms on Au(111) are prone to intermix with Au adatoms, giving rise to the Ce-Au heterometallic clusters that play as the coordination centers in the multinuclear coordination structures. The discussion is added to the manuscript as follow:

Last paragraph, Page 11: “First of all, the Ce-Au binary surface system in the absence of molecular ligands is considered. It is documented that the preparation of 2D LnAu₂ intermetallic compounds has been achieved for a number of Ln metals, namely, La³², Ce^{32,33}, Gd^{34,35,36}, Tb^{37,38}, Ho³⁷ and Er³⁷, by the surface alloying of Ln with the Au(111) substrate. All these LnAu₂ films share the same lattice structure in which each pair of the nearest-neighboring Ln atoms are bridged by two Au atoms^{32,34,35,37}. Charge transfer from Ln to Au was found in these systems³⁴, which is in accordance with the remarkable electronegativity difference between Ln and Au: Ln metals are highly electropositive while Au is the most electronegative metal on the Pauling scale³⁹. The electronegativity distinction and the charge transfer are supposed to facilitate the Ln-Au intermixing but suppress the direct Ln-Ln bonding in the binary systems. Similar to Ln elements, other electropositive metals, e.g., the alkali metals Na^{40,41}, K⁴² and Cs^{43,44}, have also been reported to form surface alloys with the Au substrates. Therefore, it can be summarized from the literature that the atoms of the electropositive metals, such as Ce, usually intermix with Au atoms rather than directly bond with each other in the binary surface systems.”

2. Additional XPS and STM experiments

To obtain more experimental evidence for the existence of Au adatoms in the multinuclear clusters observed in this work, we have conducted additional XPS and STM experiments, as shown below:

(1) We carried out XPS experiments for the samples with Ce and ligand **2** (structure shown in Fig.

R1b) on Au(111). It is found that the Ce 3d spectra of the samples with different Ce-involved coordination structures exhibit two peaks characteristic of the Ce-Au alloy (*J. Phys. Chem. C* **2007**, *111*, 3685; *Surf. Sci.* **2007**, *601*, 2445). The results indicate the Ce-Au heterometallic nature of the coordinated clusters. The results and discussion are added to the main text and Supplementary Information as follows:

Paragraph 2, Page 13: “Furthermore, in order to figure out the chemical state of the Ce atoms in the coordinated clusters, X-ray photoelectron spectroscopy (XPS) measurements were carried out for the samples with Ce and ligand **2** on Au(111). Different molecule-to-Ce ratios and annealing temperatures were employed for preparing the samples to reproduce the different coordination structures. We found shift of C 1s signal toward higher binding energy by increasing the coverage of Ce (Supplementary Fig. 10a), which is attributed to the charge transfer from the ligands to the Ce atoms caused by the coordination between them. The Ce 3d spectra exhibit two broad peaks (binding energies at ~ 885 and 904 eV) that show negligible shifts with the varied molecule-to-Ce ratios and different annealing temperatures of the samples (Supplementary Fig. 10b). The two peaks are characteristic of Ce alloying with Au, as demonstrated by the XPS studies of the Ce-Au surface alloy grown on Au(111) in this work (bottom curve in Supplementary Fig. 10b) and the previous reports^{33,45}. The fact that the positions of the Ce 3d peaks are independent from the preparation conditions of the samples suggests the similar chemical states of the Ce atoms in the coordination structures that comprise metallic centers with different nuclearities. This is in agreement with the theoretical findings showing the similar charges carried by the Ce atoms in the different coordination centers (about +2 |e| per Ce atom), as described in detail below. It can be concluded from the XPS results that the coordinated clusters are composed of intermixed Ce and Au atoms.”

Supplementary Figure 10, Page S9, Supplementary Information:

Supplementary Figure 10. XPS results of the samples with Ce and ligand **2** on Au(111). (a) C 1s spectra of the samples with different molecule-to-Ce (L/Ce) ratios obtained after annealing the samples at room temperature (RT). (b) Ce 3d spectra collected by varying the molecule-to-Ce ratios and annealing temperatures of the samples. The sample of Ce-Au alloy on Au(111) was prepared by depositing Ce onto the Au(111) substrate at RT following Supplementary Refs. 1 and 2.

(2) We prepared large Ce-containing clusters comprising seven or more Ce atoms by the coordination of Ce with the pyridine ligand **3** (structure shown in Fig. R1c) on Au(111). The multinuclear clusters show similarity in structure with the CeAu₂ surface intermetallic compound grown on Au(111) (*J. Phys. Chem. C* **2007**, *111*, 3685; *Phys. Rev. B* **2013**, *88*, 125405). This comparison provides clues for identifying the multi-Ce-containing centers in the ordered coordination structures as the Ce-Au heterometallic clusters. The results and discussion are added to the main text and Supplementary Information as follows:

Paragraph 2, Page 12: “We conducted STM characterization of the large coordinated clusters with high nuclearities prepared by depositing Ce and ligand **3** on Au(111) followed by 530 K annealing, aiming at comparing the large clusters with the Ce-Au surface intermetallic compound. As shown in Supplementary Fig. 9, clusters comprising seven or more bright dots that are assigned as Ce atoms are connected by the molecular ligands to construct the less ordered coordination structure. Both Ce atoms that are coordinated with the molecules (denoted as Ce_L, examples are marked by the blue dots in Supplementary Fig. 9) and those that are surrounded by the other Ce atoms (denoted as Ce_{Ce}, examples are marked by the red dots in Supplementary Fig. 9) can be found in the multinuclear clusters. Measurement of the Ce_L-Ce_L, Ce_{Ce}-Ce_L and Ce_{Ce}-Ce_{Ce} distances leads to the similar results of $5.0 \pm 0.5 \text{ \AA}$, $5.1 \pm 0.4 \text{ \AA}$ and $5.0 \pm 0.5 \text{ \AA}$, respectively. All these values are comparable to the Ce-Ce

separation in the CeAu₂ film grown on Au(111), that is, 5.4 Å³². Moreover, the arrangement of the Ce atoms in the clusters, although distorted more or less due to the asymmetric interactions of the molecular ligands, roughly resembles the hexagonal lattice of the Ce atoms in the CeAu₂ intermetallic compound prepared on Au(111)^{32,33}. These results suggest that, on the one hand, the Ce_{Ce} atoms are highly likely to be connected with each other in the same way as that in the CeAu₂ surface alloy, i.e., to be bridged by Au adatoms, and on the other hand, the bridging Au adatoms are also responsible for the Ce_{Ce}-Ce_L and Ce_L-Ce_L connections given the similar Ce-Ce separations and arrangements of the Ce_{Ce} and Ce_L atoms in the large clusters. The analyses provide clues for identifying the two-, three- and four-Ce_L-containing clusters in the ordered coordination structures as the Ce-Au heterometallic clusters.”

Supplementary Figure 9, Page S8, Supplementary Information:

Supplementary Figure 9. STM image of the less-ordered multinuclear coordination structure formed by **3** and Ce on Au(111) ($V = 3$ mV, $I = 50$ pA, $T = 77$ K). Examples of the Ce atoms coordinated with the molecular ligands (Ce_L) and those surrounded by the other Ce atoms (Ce_{Ce}) are marked by blue and red dots, respectively. Scale bar: 1 nm.

(3) We achieved lateral tip-manipulations of the coordination structures comprising the two-Ce-containing clusters. The model without bridging Au adatoms of the coordination structure was hence excluded. The details can be found in the manuscript as follows:

Paragraph 2, Page 14: “We start from the building block of the three-lobe structure (Fig. 3b), i.e., the coordination motif consisting of a Ce dimer and six nitrile ligands **2**, to interpret the theoretical findings. Firstly, a Ce₂(CN)₆ model of the coordination motif in which the coordination center is formed by two directly approached Ce atoms was theoretically tested. ... Moreover, one can see the slight lift-up of the substrate Au atom between the two Ce atoms (Supplementary Fig. 11a), which is supposed to suppress the lateral movement of the coordinated Ce dimer since such a process requires

frequent reconstruction of the substrate. However, in contrast to the calculation results, the tip-manipulation experiment shows the lateral displacements of the coordination motif formed by a Ce dimer and six nitrile ligands as a whole (see Supplementary Fig. 12 and Supplementary Discussion for details), and hence rejects the $\text{Ce}_2(\text{CN})_6$ model.”

Paragraph 1, Page S13, Supplementary Information:

“Tip-manipulation experiments.

4-Cyano-*m*-terphenyl (denoted as **2'**, Supplementary Fig. 12a) was employed as the ligand to construct coordination structures with Ce atoms on Au(111). With the same molecular backbone as ligand **2** but only one nitrile group, ligand **2'** coordinated with Ce in the same way as that found for the three-lobe network formed by **2** and Ce, i.e., to form $\text{Ce}_2\text{Au}_2(\text{CN})_6$ motifs (Supplementary Fig. 12b) as demonstrated later in the main text, giving rise to the isolated supramolecular structures as presented in Supplementary Fig. 12c,d. The formation of the isolated supramolecules facilitates the lateral tip-manipulation experiment. As a result, sequential lateral manipulations of a coordinated supramolecule comprising a two-Ce-containing cluster were achieved as shown in Supplementary Fig. 12d-g.”

Supplementary Figure 12, Page S10, Supplementary Information:

Supplementary Figure 12. Tip-manipulation experiments. (a) Chemical structure of 4-cyano-*m*-terphenyl (**2'**). (b) Schematic model of the $\text{Ce}_2\text{Au}_2(\text{CN})_6$ motif. (c) STM image of the two-Ce-containing center of the coordinated supramolecular structure formed by **2'** and Ce on Au(111) ($V = 10$ mV, $I = 200$ pA, $T = 4.3$ K). (d-g) STM images of the same area showing the lateral displacements of the coordinated supramolecule by the sequential tip-manipulations ($V = 10$ mV, $I = 120, 30, 50, 120$ pA, respectively, $T = 4.3$ K). The blue arrows mark the paths of the tip. Scale bars: (c) 0.5 nm, (d-g) 1 nm.

3. Improvement in theoretical calculations

In order to make our demonstration of the existence of Au adatoms in the Ce-containing clusters based on the DFT calculations more convincing and reliable, we have improved our calculations in the following three aspects.

(1) We have employed a stricter convergence criterion of $0.02 \text{ eV } \text{\AA}^{-1}$ and a potential of Ce with 12 valence electrons including the 4f electron for the calculation following the reviewer's comments. All the theoretical results are updated accordingly in the revised manuscript and are supposed to be more reliable.

(2) We have conducted additional calculations as listed below:

a) The coordination model without bridging Au adatoms proposed as the building block of the three-lobe structure, i.e., the $\text{Ce}_2(\text{CN})_6$ structure, was theoretically analyzed. The calculation of the Bader charges, Ce-Ce binding energy and differential charge density of the structure reveal the repulsive Ce-Ce interaction and the reduced stability of the coordination motif due to the direct approaching of the Ce atoms. The theoretical results and discussion are added to the main text and Supplementary Information as follows:

Paragraph 2, Page 14: "Firstly, a $\text{Ce}_2(\text{CN})_6$ model of the coordination motif in which the coordination center is formed by two directly approached Ce atoms was theoretically tested. The optimized structure of $\text{Ce}_2(\text{CN})_6$ in which the simplified nitrile ligands were employed is displayed in Supplementary Fig. 11a. The calculation reveals a positive charge of $+1.98 |e|$ for each Ce atom due to the charge transfer between the Ce atoms and the Au substrate. The approaching of the two positively charged Ce atoms thus leads to a repulsive interaction between them, which is responsible for the positive (i.e., unstable) Ce-Ce binding energy of 0.30 eV , and the electron deficiency between the two Ce atoms as seen in the differential electron density map (Supplementary Fig. 11b). These results exhibit the reduced stability of the structure caused by the direct approaching of the two Ce atoms."

Supplementary Figure 11, Page S9, Supplementary Information:

Supplementary Figure 11. Theoretical results of the $\text{Ce}_2(\text{CN})_6$ structure. (a) Optimized model (upper: top view, lower: side view), (b) cross section of the differential electron density along the black dashed line in (a), and (c) simulated STM image of the $\text{Ce}_2(\text{CN})_6$ motif on Au(111).

b) The calculation of $\text{Ce}_2\text{Au}(\text{CN})_6$, another candidate model of the building block of the three-lobe structure in addition to $\text{Ce}_2(\text{CN})_6$ and $\text{Ce}_2\text{Au}_2(\text{CN})_6$, was carried out. The results show a lower stability of $\text{Ce}_2\text{Au}(\text{CN})_6$ than $\text{Ce}_2\text{Au}_2(\text{CN})_6$, which supports the identification of the $\text{Ce}_2\text{Au}_2(\text{CN})_6$ motif as the building block of the three-lobe structure. The details are added to the manuscript as follows:

Paragraph 2, Page 15: “Subsequently, two models with one or two bridging Au adatom(s) located between the two Ce atoms, i.e., $\text{Ce}_2\text{Au}(\text{CN})_6$ and $\text{Ce}_2\text{Au}_2(\text{CN})_6$, respectively, were tested. The atomic arrangement of the Ce_2Au_2 cluster in the latter model is proposed according to the lattice structure of the CeAu_2 surface intermetallic compound grown on Au(111)³². The optimized models of the two candidate structures are displayed in Supplementary Fig. 13a and Fig. 5a, respectively. The calculations reveal that the energy of the $\text{Ce}_2\text{Au}_2(\text{CN})_6$ [$\text{Ce}_2\text{Au}(\text{CN})_6$] structure is 1.57 eV (0.75 eV) lower (i.e., more stable) than the total energy of the $\text{Ce}_2(\text{CN})_6$ motif plus two (one) non-interacting Au adatom(s). ... Given that Au adatoms generated from the step and diffusing on the terrace should be well accessible on Au(111) under the reaction conditions⁴⁶⁻⁵⁰, we propose the most energetically preferred $\text{Ce}_2\text{Au}_2(\text{CN})_6$ structure among the proposed models as the building block of the three-lobe structure, as illustrated by the superimposed molecular models in Fig. 3b.”

Supplementary Figure 13, Page S11, Supplementary Information:

Supplementary Figure 13. Theoretical results of the $\text{Ce}_2\text{Au}(\text{CN})_6$ structure. (a) Optimized model (upper: top view, lower: side view) and (b) simulated STM image of the $\text{Ce}_2\text{Au}(\text{CN})_6$ motif on Au(111).

c) We calculated the simulated STM images of the $\text{Ce}_2(\text{CN})_6$ and $\text{Ce}_2\text{Au}(\text{CN})_6$ structures. The resulted STM simulations were compared with that of $\text{Ce}_2\text{Au}_2(\text{CN})_6$. As a result, the simulated STM image of $\text{Ce}_2\text{Au}_2(\text{CN})_6$ shows the best agreement with the experimental data. The simulated STM images are added to the Supplementary Information as Supplementary Figs. 11c and 13b, respectively, as displayed above. The discussion is added to the main text as follow:

Paragraph 1, Page 16: “Moreover, the simulated STM image of the optimized $\text{Ce}_2\text{Au}_2(\text{CN})_6$ structure (Fig. 5b) was compared with those of $\text{Ce}_2(\text{CN})_6$ (Supplementary Fig. 11c) and $\text{Ce}_2\text{Au}(\text{CN})_6$

(Supplementary Fig. 13b). As a result, the STM simulation of $\text{Ce}_2\text{Au}_2(\text{CN})_6$ shows the best agreement with the experimental data, which supports the identification of the $\text{Ce}_2\text{Au}_2(\text{CN})_6$ structure as the coordination motif to construct the three-lobe network.”

(3) We have compared the Ce-Ce distances achieved by theoretical optimization with the experimental values, showing the agreement between the calculated and experimental results. The details can be found in the main text as follows:

Paragraph 2, Page 15: “The Ce-Ce distance in the optimized model of the $\text{Ce}_2\text{Au}_2(\text{CN})_6$ motif (4.95 Å) is consistent with the experimental value (4.7 ± 0.4 Å) by taking the measurement error into account.”

Paragraph 2, Page 19: “The Ce-Ce distances in these models, i.e., 4.52 Å for $\text{Ce}_2\text{Au}_2\text{Py}_4$, 5.56 Å for $\text{Ce}_3\text{Au}_4\text{Py}_6$ and 4.77 Å for $\text{Ce}_4\text{Au}_5(\text{CN})_8$, are consistent with the experimentally measured values, that is, 4.5 ± 0.3 Å, 5.1 ± 0.5 Å and 4.9 ± 0.5 Å, respectively.”

Based on the theoretical results presented in the revised manuscript, the existence of Au adatoms in the coordinated clusters can be demonstrated in the following aspects in addition to the total energy calculations:

- (1) The repulsive Ce-Ce interaction and the reduced stability of the coordination model without Au adatoms.
- (2) The better agreement with the experimental data of the simulated STM image of the model with two Au adatoms than those without.
- (3) The consistency in the Ce-Ce distances between the proposed models comprising Au adatoms and the experimentally observed structures.

Comment:

... Furthermore, no discussion about a possible mechanism for the formation of this unusual complex is given.

Author reply:

We have added three paragraphs to the main text to discuss the mechanism for the formation of the multinuclear coordination structures as follow:

Pages 21 and 22: “Next, we discuss the possible mechanism for the formation of the multinuclear coordination structures. All the multinuclear coordination structures achieved in this work feature relatively low molecule-to-Ln ratios (no more than 3:2). Each Ce atom in these structures coordinates with 2 or 3 molecular ligands. As a comparison, relatively high molecule-to-Ln (or molecular coordination site-to-Ln in the case of chelating ligands being engaged) ratios (3:2 or higher) were

frequently found for the previously reported Ln-engaged mononuclear coordination structures constructed on surfaces¹⁴⁻²⁴, in which the single Ln atoms afforded multi (≥ 3)-fold coordination with the molecular ligands. Accordingly, we propose that the low molecule-to-Ln ratios employed for the preparation of the metal-organic hybrids in this work **should play a key role in the formation of the multinuclear coordination structures.**

One may expect two types of coordination products in the samples with a molecule-to-Ln ratio $\leq 3:2$, i.e., the mononuclear structures with low coordination numbers (namely, two- or three-fold coordination), and the multinuclear structures. The comparison of the stability of these two types of coordination structures can be carried out based on the calculation results shown above. For the two-fold and three-fold mononuclear structures, the stabilizing energy per Ce atom due to the formation of the coordination bonds with the ligands can be estimated as about -2.6 eV and -3.9 eV, respectively, by taking the averaged Ce-N binding energy of the coordination structures concerned in this work for evaluation. As for the multinuclear structures constructed by the $\text{Ce}_2\text{Au}_2\text{Py}_4$, $\text{Ce}_3\text{Au}_4\text{Py}_6$, $\text{Ce}_2\text{Au}_2(\text{CN})_6$ or $\text{Ce}_4\text{Au}_5(\text{CN})_8$ motifs, the stabilizing energy per Ce atom contributed by the bonding with both the molecular ligands and the Au adatoms ranges from -4.4 to -5.4 eV. As a conclusion, the multinuclear coordination structures are thermodynamically preferred at low molecule-to-Ln ratios. Thus, the reduction in the molecule-to-Ln ratio of the sample, which can be achieved by either deposition of additional Ce or molecular desorption caused by thermal treatment of the sample, can serve as a driving force for the emergence of the multinuclear coordination structures.

Moreover, thermal treatment of the sample is also necessary for the preparation of the multinuclear structures. In addition to tuning the molecule-to-Ln ratio by the thermal-induced molecular desorption as mentioned above, the thermal treatment also provides energies for the diffusion and rearrangement of the molecules and metal adatoms on the surface, which is indispensable for obtaining the thermodynamically preferred multinuclear products. Otherwise, coordination structures with lower stability may emerge, such as the three-fold mononuclear structures (namely, the single-wall honeycomb network formed by **1** and the Sierpiński triangles formed by **2**) obtained at room temperature in this work.”

Comment:

The computational method section also raises several questions. The authors claim to have used PBE as density functional and treat dispersion interaction by optB88. This does not make sense since the latter is a vdW-density functional which is used standalone and is not combined with PBE. Thus, it is unclear how the numbers in the manuscript are derived. It also does not lend confidence in the computational section altogether.

Author reply:

We thank the reviewer for pointing out this problem. The functional employed in this work is opt-B88 while PBE was not used. We have revised the misleading description of density functional in the Methods section as follow:

Paragraph 2, Page 24: “The exchange-correlation energy was calculated with the opt-B88 functional in order to take the van der Waals (vdW) dispersive correction into consideration⁵⁵⁻⁵⁷.”

Comment:

The question if the 4f electrons of Ce are really not participating in the bonding (as implicitly assumed by using a large core pseudopotential) also needs investigation.

Author reply:

We have updated all the theoretical results by using a potential of Ce with 12 valence electrons including the 4f electron. It is found that the previous conclusions are still solid based on the updated calculation results. The description of the Ce potential used in this work has been revised accordingly as follow:

Paragraph 2, Page 24: “The potential of Ce was generated with 12 valence electrons including the 4f electron.”

Comment:

Plotting the difference density w.r.t. the atomic densities leads to a picture where many aspects are mixed. Thus, it is not clear at all where the density change comes from: bonding or polarization. The pictures should be plotted w.r.t. the free molecule and the free surface which is the standard procedure in the field (and even then it is often not clear what one sees).

Author reply:

We fully agree with the reviewer that to calculate the electron density difference with respect to the electron densities of the constituents of the system (the molecules, substrate, etc.) is the standard procedure in the field. In fact, it is exactly how we plotted the differential electron density maps in this work.

The method suggested by the reviewer has been frequently adopted in a number of previous works for calculation of electron density difference of the on-surface metal-organic hybrid structures, aiming at figuring out the interactions between the constituents in the systems (to name a few, *Nano Lett.* **2007**, 7, 3813; *J. Am. Chem. Soc.* **2011**, 133, 12031; *Nat. Commun.* **2012**, 3, 940; *Small* **2015**, 11, 6358; *Chem. Commun.* **2016**, 52,12944; *Angew. Chem. Int. Ed.* **2018**, 57, 4617). In a recent work by Yan et al.

(*Angew. Chem. Int. Ed.* **2018**, *57*, 4617), such a method was used to elucidate the electron density rearrangement in the surface-confined coordinated heterometallic cluster structures which is quite similar to the system studied in this work. Following the literature, we calculated the differential electron density as the subtraction between the electron density of the total interacting system and the sum of that of its constituents, i.e., the ligand molecules, substrate, Ce adatoms, and Au adatoms. The ligand molecules and substrate, respectively, are taken as a whole for the calculation of their electron densities, while the Ce and Au adatoms are treated separately since what is concerned here is the electron density rearrangement due to the Ce-Au and Ce-ligand interactions. Therefore, the differential electron density maps presented in this work are plotted with respect to the electron densities of the molecules, substrate and metal adatoms involved in the coordination, as suggested by the reviewer.

Comment:

What has not been discussed at all is how the optimized structures were derived. Have several reasonable starting structures been assumed? How was the Ce-Ce distance chosen on the surface (which seems very large in Figure S6)?

Author reply:

For the building block of the three-lobe structure (Fig. 3b), i.e., the coordination motif consisting of a Ce dimer and six nitrile ligands **2** (structure shown in Fig. R1b), we tested three candidate models in which the coordination center is formed by two Ce atoms bridged by two $[\text{Ce}_2\text{Au}_2(\text{CN})_6]$, one $[\text{Ce}_2\text{Au}(\text{CN})_6]$, or no $[\text{Ce}_2(\text{CN})_6]$ Au adatom(s), respectively. The initial structural parameters, including the Ce-Ce distances, of the three candidate models were set as follows:

(1) $\text{Ce}_2\text{Au}_2(\text{CN})_6$: The arrangement of the Ce and Au atoms as well as the Ce-Ce distance in the Ce_2Au_2 cluster were set according to the lattice structure of the CeAu_2 intermetallic compound prepared on Au(111) (*J. Phys. Chem. C* **2007**, *111*, 3685; *Phys. Rev. B* **2013**, *88*, 125405).

(2) $\text{Ce}_2\text{Au}(\text{CN})_6$: The initial structure was obtained by removing a Au adatom from the optimized model of $\text{Ce}_2\text{Au}_2(\text{CN})_6$.

(3) $\text{Ce}_2(\text{CN})_6$: Initial inputs with different Ce-Ce separations (2.5 Å, 3.0 Å and 3.6 Å) were tested, all of which converged to the same final structure as shown in Supplementary Fig. 11a. The large Ce-Ce distance in the optimized model of $\text{Ce}_2(\text{CN})_6$ can be understood as the result of the repulsive interaction between the two positively charged Ce atoms (+1.98 |e| per Ce atom) which is due to the charge transfer between the Ce atoms and the Au substrate. In fact, the energy calculation reveals a positive Ce-Ce binding energy of 0.30 eV in $\text{Ce}_2(\text{CN})_6$, indicating that the approaching of the two Ce atoms reduces the stability of the structure, which also explains the large Ce-Ce separation.

The optimized models of $\text{Ce}_2(\text{CN})_6$, $\text{Ce}_2\text{Au}(\text{CN})_6$ and $\text{Ce}_2\text{Au}_2(\text{CN})_6$ are presented in Supplementary Figs. 11a, 13a (displayed in our replies above) and Fig. 5a, respectively. The $\text{Ce}_2\text{Au}(\text{CN})_6$ and $\text{Ce}_2(\text{CN})_6$ structures were ruled out according to the theoretical analyses of the optimized models. The details have been added to the main text as follow:

Pages 14-16: “The existence of bridging Au adatoms in the coordinated clusters is also supported by the calculation results. We start from the building block of the three-lobe structure (Fig. 3b), i.e., the coordination motif consisting of a Ce dimer and six nitrile ligands **2**, to interpret the theoretical findings. Firstly, a $\text{Ce}_2(\text{CN})_6$ model of the coordination motif in which the coordination center is formed by two directly approached Ce atoms was theoretically tested. The optimized structure of $\text{Ce}_2(\text{CN})_6$ in which the simplified nitrile ligands were employed is displayed in Supplementary Fig. 11a. The calculation reveals a positive charge of +1.98 |e| for each Ce atom due to the charge transfer between the Ce atoms and the Au substrate. The approaching of the two positively charged Ce atoms thus leads to a repulsive interaction between them, which is responsible for the positive (i.e., unstable) Ce-Ce binding energy of 0.30 eV, and the electron deficiency between the two Ce atoms as seen in the differential electron density map (Supplementary Fig. 11b). These results exhibit the reduced stability of the structure caused by the direct approaching of the two Ce atoms. Moreover, one can see the slight lift-up of the substrate Au atom between the two Ce atoms (Supplementary Fig. 11a), which is supposed to suppress the lateral movement of the coordinated Ce dimer since such a process requires frequent reconstruction of the substrate. However, in contrast to the calculation results, the tip-manipulation experiment shows the lateral displacements of the coordination motif formed by a Ce dimer and six nitrile ligands as a whole (see Supplementary Fig. 12 and Supplementary Discussion for details), and hence rejects the $\text{Ce}_2(\text{CN})_6$ model.

Subsequently, two models with one or two bridging Au adatom(s) located between the two Ce atoms, i.e., $\text{Ce}_2\text{Au}(\text{CN})_6$ and $\text{Ce}_2\text{Au}_2(\text{CN})_6$, respectively, were tested. The atomic arrangement of the Ce_2Au_2 cluster in the latter model is proposed according to the lattice structure of the CeAu_2 surface intermetallic compound grown on Au(111)³². The optimized models of the two candidate structures are displayed in Supplementary Fig. 13a and Fig. 5a, respectively. The calculations reveal that the energy of the $\text{Ce}_2\text{Au}_2(\text{CN})_6$ [$\text{Ce}_2\text{Au}(\text{CN})_6$] structure is 1.57 eV (0.75 eV) lower (i.e., more stable) than the total energy of the $\text{Ce}_2(\text{CN})_6$ motif plus two (one) non-interacting Au adatom(s). Note that the Au adatom absorbed on the terrace of the substrate is taken as the reference state for the energy calculation here, whereas the Au adatoms on the Au substrate are usually stabilized by re-attaching to the step edges in the real conditions. In the latter context, the energy differences between the proposed structures with and without the bridging Au adatom(s) can be estimated by subtracting the formation energy of the adatom(s) emitted from the step (0.500 eV per Au adatom⁴⁶) from the above-presented

results. This gives rise to the energy differences of 0.57 eV and 0.25 eV for $\text{Ce}_2\text{Au}_2(\text{CN})_6$ and $\text{Ce}_2\text{Au}(\text{CN})_6$, respectively, lower than $\text{Ce}_2(\text{CN})_6$, which still indicates an energetic preference for formation of the structures with the bridging Au adatom(s). Given that Au adatoms generated from the step and diffusing on the terrace should be well accessible on Au(111) under the reaction conditions⁴⁶⁻⁵⁰, we propose the most energetically preferred $\text{Ce}_2\text{Au}_2(\text{CN})_6$ structure among the proposed models as the building block of the three-lobe structure, as illustrated by the superimposed molecular models in Fig. 3b. The similar phenomenon was also found by Yan et al.³⁰ showing that the evaporated metal atoms were bridged by the intrinsic adatoms to form coordinated heterometallic clusters on surfaces. The Ce-Ce distance in the optimized model of the $\text{Ce}_2\text{Au}_2(\text{CN})_6$ motif (4.95 Å) is consistent with the experimental value (4.7 ± 0.4 Å) by taking the measurement error into account. Moreover, the simulated STM image of the optimized $\text{Ce}_2\text{Au}_2(\text{CN})_6$ structure (Fig. 5b) was compared with those of $\text{Ce}_2(\text{CN})_6$ (Supplementary Fig. 11c) and $\text{Ce}_2\text{Au}(\text{CN})_6$ (Supplementary Fig. 13b). As a result, the STM simulation of $\text{Ce}_2\text{Au}_2(\text{CN})_6$ shows the best agreement with the experimental data, which supports the identification of the $\text{Ce}_2\text{Au}_2(\text{CN})_6$ structure as the coordination motif to construct the three-lobe network. In both experimental images and STM simulation of $\text{Ce}_2\text{Au}_2(\text{CN})_6$, the two Ce atoms are resolved as the two bright dots while the Au adatoms are invisible. This can be understood by the larger atomic radius of Ce than Au which leads to a larger height in z direction of the former than the latter when they are adsorbed on the Au substrate, as displayed by the side view of the optimized model (Fig. 5a bottom). The similar situation that only the metal atoms with a bigger size in the heterometallic clusters are discernable by STM was also reported by Yan et al. recently³⁰.

Fig. 5. Theoretical results of the $\text{Ce}_2\text{Au}_2(\text{CN})_6$ structure. (a) DFT optimized models (upper: top view, lower: side view), (b) simulated STM image and (c) cross section of the differential electron density along the black dashed line in (a) of the $\text{Ce}_2\text{Au}_2(\text{CN})_6$ motif on Au(111). Simplified molecular ligands are employed.”

The structure of the Ce_2Au_2 center in $\text{Ce}_2\text{Au}_2(\text{CN})_6$ resembles the atomic arrangement found in the CeAu_2 surface intermetallic compound prepared on Au(111), that is, each pair of the nearest-

neighboring Ce atoms are bridged by two Au atoms (*Phys. Rev. B* **2013**, 88, 125405). Therefore, the structures of the coordinated clusters in all the other multinuclear coordination structures were also proposed according to the lattice structure of the CeAu₂ film. The so-achieved optimized models of the coordination motifs are presented in Fig. 6a-c, showing the Ce₂Au₂Py₄ motifs as the building block of the Kagome (Figs. 2b and 4a), fishing-net (Fig. 4b) and chain-like (Fig. 4c) structures, while the double-wall honeycomb (Fig. 2c) and four-lobe (Fig. 3c) structures are constructed by the Ce₃Au₄Py₆ and Ce₄Au₅(CN)₈ motifs, respectively. The statements about the Ce₂Au₂Py₄, Ce₃Au₄Py₆ and Ce₄Au₅(CN)₈ motifs and their optimized models have been added to the main text as follows:

Paragraph 2, Page 19: “Given the critical role of the bridging Au adatoms in stabilizing the Ce₂Au₂(CN)₆ structure, we argue that the coordination centers of all the other multinuclear coordination structures achieved in this work are composed of the Ce-Au heterometallic clusters in which each pair of the nearest-neighboring Ce atoms are bridged by two Au adatoms, as proposed according to the atomic arrangement of the CeAu₂ surface alloy prepared on Au(111)³² (see Supplementary Fig. 14 and Supplementary Discussion for other tested models of the coordinated four-Ce-containing cluster). The optimized models are presented in Fig. 6a-c, showing the Ce₂Au₂Py₄ motif as the building block of the Kagome (Figs. 2b and 4a), fishing-net (Fig. 4b) and chain-like (Fig. 4c) structures, while the double-wall honeycomb (Fig. 2c) and four-lobe (Fig. 3c) structures are constructed by the Ce₃Au₄Py₆ and Ce₄Au₅(CN)₈ motifs, respectively, as illustrated by the molecular models superimposed in Figs. 2b, 2c, 3c and 4.”

Figure 6a-c, Page 20:

Fig. 6. Theoretical results of the multinuclear coordination structures. Optimized models of (a) the Ce₂Au₂Py₄, (b) the Ce₃Au₄Py₆ and (c) the Ce₄Au₅(CN)₈ motifs (upper: top view, lower: side view). ...

Moreover, for the coordination center of the four-lobe network (Fig. 3c), i.e., the four-Ce-containing cluster, we also tested four other starting structures in addition to the Ce₄Au₅ cluster in the

Ce₄Au₅(CN)₈ motif as shown in Fig. 6c in the main text. The Ce-Ce and Ce-Au distances in the initial state of these models were set by referring to the corresponding parameters in the optimized models of the Ce₂Au₂(CN)₆, Ce₂Au₂Py₄ and Ce₃Au₄Py₆ structures. The optimized models are presented in Supplementary Fig. 14. All these four structures were rejected due to the inconsistency of their geometries with the experimental observations. The calculation results and discussion of the four tested models of the Ce tetramer are added to Supplementary Information as follows:

Paragraph 2, Page S13, Supplementary Information:

“Theoretically tested models of the coordinated four-Ce-containing cluster.

For the coordination structure comprising Ce tetramers, i.e., the four-lobe network, in addition to the Ce₄Au₅(CN)₈ motif as shown in Fig. 6c in the main text, several other structures were also tested. Firstly, three candidate models of the Ce-Au cluster other than Ce₄Au₅ were tested without taking the molecular ligands into account. They are two Ce₄Au₄ clusters in which the four Au adatoms are located at either the inner (denoted as Ce₄Au₄-*in*) or the outer (denoted as Ce₄Au₄-*out*) side of the four Ce atoms, and a Ce₄Au₆ cluster. The optimized model of Ce₄Au₄-*in* (Supplementary Fig. 14a) shows a Ce-Ce distance of 5.66 Å, which is larger than the experimentally measured value (4.9 ± 0.5 Å). The Ce atoms in the optimized models of Ce₄Au₄-*out* (Supplementary Fig. 14b) and Ce₄Au₆ (Supplementary Fig. 14c) are arranged in the different symmetries from the experimentally observed square-shaped Ce tetramers. As a consequence, all the three structures were ruled out due to their poor consistency with the experimental results. Moreover, the Ce₄Au₅ cluster located at a different position with respect to the substrate from that presented in the main text (Fig. 6c) was also considered (denoted as Ce₄Au₅-*top* since the central Au adatom of the cluster was located at the top site of the substrate in this model). The optimized model of the Ce₄Au₅-*top* cluster is shown in Supplementary Fig. 14d. The further optimization of the coordination structure constructed by adding eight nitrile ligands to the optimized Ce₄Au₅-*top* cluster resulted in the distortion of the metallic center (Supplementary Fig. 14e), which disagrees with the experimental observations. Therefore, the Ce₄Au₅-*top*-based coordination motif was also excluded as the building block of the four-lobe network.”

Supplementary Figure 14, Page S12, Supplementary Information:

Supplementary Figure 14. Theoretically tested models of the four-Ce-containing cluster. Optimized models of (a) the $\text{Ce}_4\text{Au}_4\text{-in}$ cluster, (b) the $\text{Ce}_4\text{Au}_4\text{-out}$ cluster, (c) the Ce_4Au_6 cluster, (d) the $\text{Ce}_4\text{Au}_5\text{-top}$ cluster, and (e) the $\text{Ce}_4\text{Au}_6(\text{CN})_8$ motif constructed based on the $\text{Ce}_4\text{Au}_5\text{-top}$ cluster on Au(111) (upper: top view, lower: side view).

A sentence has been added to the Calculation Methods section in the main text as follow to interpret how the initial geometries of the coordinated clusters in the models for optimization were set:

Paragraph 2, Page 24: “The initial geometries of the Ce-Au clusters in the models for optimization were set either according to the lattice structure of the CeAu_2 intermetallic compound prepared on Au(111)³², or by referring to the parameters achieved in other optimized structures studied in this work.”

Comment:

Also, the convergence criterion is not strict enough ($<0.02 \text{ eV/\text{Ang}}$).

Author reply:

We have updated all the theoretical results by using a stricter convergence criterion of 0.02 eV \AA^{-1} following the reviewer’s comment. Such a convergence criterion has been frequently used for the calculations of on-surface molecular systems (to name a few, *ACS Nano* **2014**, 8, 11799; *ACS Nano*

2014, 8, 8644; ACS Nano 2017, 11, 8302). The description of the convergence criterion in the Methods section on Page 24 has been revised accordingly.

Comment:

I see the need for direct experimental evidence for the proposed structures by revealing the nature of the atoms in the on-surface structure. This could for example be derived by XPS or NEXAFS investigations. Up to now, the structural determination relies too much on interpretation of (partially weak) indirect evidence.

Author reply:

We appreciate the reviewer for the helpful comment. In order to make our conclusions more solid, we have made major revisions to the manuscript in the following three aspects:

Firstly, we have added discussion based on literature research;

Secondly, we have conducted additional XPS and STM experiments;

Thirdly, we have updated the theoretical analyses by improving the calculation methods, adding new calculation results, and comparing the calculation results with the experimental data.

Having added these new discussion and results, we rewrote the section of “Structure of the Coordinated Multi-Ce-Containing Clusters” on Pages 11-21 in the main text to interpret the structures of the coordinated multi-Ce-containing clusters. The details of the revisions can be found in our replies above. We believe that the so-revised manuscript provides more evidence to confirm the proposed models of the coordination structures, showing that Au adatoms are indispensable for stabilizing the multi-Ce-containing coordination centers.

Reviewer #2

Comment:

Wang and coworkers report an elegant approach for the controlled on-surface formation of ligand-stabilized heterometallic Ce/Au clusters with two, three and four Ce atoms. The different cluster sizes are realized by variation of the ligand structure (nitrile and pyridine based) and the preparation procedures, resulting in different ligand-to-metal ratios. The diversity of the structures and the level of control are unprecedented and very impressive. The high-quality STM data clearly show the positions of the Ce atoms and the ligands. The presence of additional Au atoms in the clusters is derived from plausibility considerations and is supported by DFT calculations. The DFT calculations also reveal that the Ce-Au bond has partial ionic character, owing to the large electronegativity difference. This

work is highly original and carefully composed. Publication in *Nature Communications* is recommended after some minor issues have been addressed.

1. On page 10, it is stated that the $Ce_2Au_2L_6$ structure ($L = \text{ligand}$) is lower in energy than the Ce_2L_6 structure plus two non-interacting Au adatoms by 1.94 eV. From this, it is concluded that the $Ce_2Au_2L_6$ structure is energetically stabilized. However, to make this conclusion, the non-interacting Au adatoms are not the correct point of reference: If the Au adatoms gain even more energy by re-attaching to a step edge (where they typically originate from), then the $Ce_2Au_2L_6$ structure would not be stabilized. This can easily be understood by considering that the equilibrium concentration of adatoms decreases when the temperature is lowered. When the adatoms return to the step edges, also the $Ce_2Au_2L_6$ structure would break apart if the Au adatoms gain more energy by binding to the step edge. (Unless $Ce_2Au_2L_6$ is a kinetically trapped state, but this is not what the authors mean.) Therefore, the energy gain must be calculated relative to the Ce_2L_6 structure plus two Au atoms at a step edge. A typical literature value for the energy of formation of an Au adatom on Au(111) is 0.61 eV. With this value, the energy gain by formation of the $Ce_2Au_2L_6$ structure would be reduced to 0.72 eV, which still represents a considerable stabilization.

Author reply:

We thank the reviewer for the helpful comment. In our calculation, by taking the Au adatoms absorbed on the terrace of the substrate as the reference state, the resulted energy difference between the $Ce_2Au_2(CN)_6$ and $Ce_2(CN)_6$ structures (1.57 eV based on the updated calculation methods) reflects the different stabilities of the two proposed structures on an ideal, defect-free Au(111) substrate. However, as commented by the reviewer, the Au adatoms are usually stabilized by re-attaching to the step edges in the real conditions. In this context, the energy difference between the two proposed structures can be estimated by subtracting the formation energy of the adatoms generated from the step from the above-mentioned result. The formation energy of a Au adatom on Au(111) ranges from 0.50 to 0.79 eV according to the literature (*J. Phys.: Condens. Matter* **1994**, 6, 9495; *Surf. Sci.* **2006**, 600, 1277; *J. Phys. Chem. C* **2007**, 111, 12149; *Nanoscale*, **2014**, 6, 10850; *J. Phys. Chem. C* **2020**, 124, 7492) due to the varied theoretical methods applied and the different situations concerned (e.g. generation of an adatom from a terrace, step or kink). A formation energy of 0.50 eV reported for an adatom emitted from a step on Au(111) (*J. Phys.: Condens. Matter* **1994**, 6, 9495) is used here. As a result, the value of the energy difference reduces to 0.57 eV, which still represents a considerable stabilization of the $Ce_2Au_2(CN)_6$ structure, as commented by the reviewer. In the revised manuscript, we have also added the calculation results of another candidate structure, $Ce_2Au(CN)_6$. The different reference states of Au adatoms were also considered for the energy calculation involving the

Ce₂Au(CN)₆ structure. These results and discussion have been added to the main text as follow:

Paragraph 2, Page 15: “The calculations reveal that the energy of the Ce₂Au₂(CN)₆ [Ce₂Au(CN)₆] structure is 1.57 eV (0.75 eV) lower (i.e., more stable) than the total energy of the Ce₂(CN)₆ motif plus two (one) non-interacting Au adatom(s). Note that the Au adatom absorbed on the terrace of the substrate is taken as the reference state for the energy calculation here, whereas the Au adatoms on the Au substrate are usually stabilized by re-attaching to the step edges in the real conditions. In the latter context, the energy differences between the proposed structures with and without the bridging Au adatom(s) can be estimated by subtracting the formation energy of the adatom(s) emitted from the step (0.500 eV per Au adatom⁴⁶) from the above-presented results. This gives rise to the energy differences of 0.57 eV and 0.25 eV for Ce₂Au₂(CN)₆ and Ce₂Au(CN)₆, respectively, lower than Ce₂(CN)₆, which still indicates an energetic preference for formation of the structures with the bridging Au adatom(s).”

Comment:

2. On pages 12 and 13, the total calculated Ce-Au binding energies in Ce₂Au₂L₆ and the other structures are presented. According to the methods section (page 20), Au adatoms are used in the reference state. While the resulting values are certainly meaningful, they do not really describe the energetic stabilization of the clusters, as explained in the previous comment. To confirm the energetic stabilization of the structures, referencing to Au atoms at step edges is again more reasonable. This would increase all energy values by around 1.2 eV, but all of them would still be negative. ...

Author reply:

We have estimated the Ce-Au binding energies of the multinuclear structures with the Au atom at the step edge on Au(111) as the reference state following the reviewer’s comment by adding the formation energy of the Au adatoms emitted from the step on Au(111) (0.500 eV per atom according to *J. Phys.: Condens. Matter* **1994**, 6, 9495) to $E_{\text{Ce-Au}}^{\text{tot}}$ as listed in Table 2 in the main text. The resulted binding energies are still negative and represent considerable stabilization effect of the Ce-Au bonds. The statement of the different reference states of Au adatoms used for the binding energy calculation has been added to the main text as follow:

Paragraph 1, Page 25: “Here, the Au adatom on the terrace of the substrate is taken as the reference state for the energy calculation. The total Ce-Au binding energy relative to the Au atom at the step edge can be estimated by adding the formation energy of the Au adatoms emitted from the step on Au(111) (0.500 eV per atom⁴⁶) to $E_{\text{Ce-Au}}^{\text{tot}}$. The so-achieved results still represent considerable stabilization effect of the Ce-Au bonds.”

Comment:

... Since the authors emphasize that the Ce-Au interactions stabilize the clusters (rather than Au-Au or Ce-Ce interactions), it may also be instructive to compare with the energies of the corresponding homometallic clusters (Au₄ and Ce₄).

Author reply:

We carried out the comparison between the heterometallic Ce₂Au₂ cluster and the corresponding homometallic Au₄ and Ce₄ clusters by using the simplified models with only the metal atoms but no molecular ligands for calculation so as to minimize the computation cost. As a consequence, the total Ce-Au binding energy in the Ce₂Au₂ cluster (optimized model shown in Fig. R2a) is -3.80 eV. As a comparison, the total Au-Au binding energy in the Au₄ cluster (optimized model shown in Fig. R2b) is -1.09 eV, showing a lower stability of the Au₄ cluster than the Ce₂Au₂ cluster. Note that the Au adatom on the terrace is taken as the reference state for the energy calculation. With the Au atom at the step edge on Au(111) as the reference state, the total binding energies would be -2.80 eV for Ce₂Au₂ and 0.91 eV for Au₄. The positive binding energy of the Au₄ cluster suggests that comparing with being engaged in the construction of the Au₄ cluster, Au adatoms on the Au(111) substrate prefer to re-attach to the step edges. As for the Ce₄ cluster, we found distorted geometries of the cluster during the optimization, suggesting the unfavorable formation of the Ce₄ cluster. This can be understood by the repulsive interaction between the positively charged Ce atoms originated from the charge transfer between the Ce atoms and the Au substrate. Further confirmation of the Ce-Ce repulsive interaction comes from the optimization of the Ce₂(CN)₆ motif (optimized model shown in Supplementary Fig. 11a) which reveals a positive binding energy of 0.30 eV between the two positively charge Ce atoms (+1.98 |e| per Ce atom).

Fig. R2. Optimized models of (a) the Ce₂Au₂ cluster and (b) the Au₄ cluster on Au(111) (upper: top view, lower: side view).

To demonstrate that it is the Ce-Au bonding rather than the homometallic interactions that stabilizes the coordinated clusters, we have added the discussion to show the preferred formation of Ln-Au bonds and the unfavorable bonding between Ln atoms in the Ln-Au bimetallic systems as follow:

Last paragraph, Page 11: “First of all, the Ce-Au binary surface system in the absence of molecular ligands is considered. It is documented that the preparation of 2D LnAu₂ intermetallic compounds has been achieved for a number of Ln metals, namely, La³², Ce^{32,33}, Gd^{34,35,36}, Tb^{37,38}, Ho³⁷ and Er³⁷, by the surface alloying of Ln with the Au(111) substrate. All these LnAu₂ films share the same lattice structure in which each pair of the nearest-neighboring Ln atoms are bridged by two Au atoms^{32,34,35,37}. Charge transfer from Ln to Au was found in these systems³⁴, which is in accordance with the remarkable electronegativity difference between Ln and Au: Ln metals are highly electropositive while Au is the most electronegative metal on the Pauling scale³⁹. The electronegativity distinction and the charge transfer are supposed to facilitate the Ln-Au intermixing but suppress the direct Ln-Ln bonding in the binary systems. Similar to Ln elements, other electropositive metals, e.g., the alkali metals Na^{40,41}, K⁴² and Cs^{43,44}, have also been reported to form surface alloys with the Au substrates. Therefore, it can be summarized from the literature that the atoms of the electropositive metals, such as Ce, usually intermix with Au atoms rather than directly bond with each other in the binary surface systems.”

Moreover, the calculation results of the Ce₂(CN)₆ motif which indicate the unstable Ce-Ce interaction in the coordination structure have been added to the main text and Supplementary Information as follows:

Paragraph 2, Page 14: “Firstly, a Ce₂(CN)₆ model of the coordination motif in which the coordination center is formed by two directly approached Ce atoms was theoretically tested. The optimized structure of Ce₂(CN)₆ in which the simplified nitrile ligands were employed is displayed in Supplementary Fig. 11a. The calculation reveals a positive charge of +1.98 |e| for each Ce atom due to the charge transfer between the Ce atoms and the Au substrate. The approaching of the two positively charged Ce atoms thus leads to a repulsive interaction between them, which is responsible for the positive (i.e., unstable) Ce-Ce binding energy of 0.30 eV, and the electron deficiency between the two Ce atoms as seen in the differential electron density map (Supplementary Fig. 11b). These results exhibit the reduced stability of the structure caused by the direct approaching of the two Ce atoms.”

Supplementary Figure 11, Page S9, Supplementary Information:

Supplementary Figure 11. Theoretical results of the $\text{Ce}_2(\text{CN})_6$ structure. (a) Optimized model (upper: top view, lower: side view), (b) cross section of the differential electron density along the black dashed line in (a), and (c) simulated STM image of the $\text{Ce}_2(\text{CN})_6$ motif on Au(111).

Comment:

3. *Methods:* Since Au and also Ce show substantial influence of relativistic effects, it should be mentioned whether a scalar relativistic (SR) or fully relativistic (FR, spin-orbit coupling included) approach was used. There can be substantial energy differences between SR and FR results (see e.g. P. Pyykkö, *Annu. Rev. Phys. Chem.* 63 (2012) 45-64).

Author reply:

The calculations in this work have taken into account the scalar relativistic effect which includes the mass-velocity and Darwin correction terms, as implemented in VASP (*J. Comput. Chem.* **2008**, 29, 2044). The similar methods are frequently employed for the theoretical investigation of the Au- or Ln-involved systems (*Chem. Phys. Lett.* **2004**, 392, 452; *Small* **2015**, 11, 6358; *J. Phys. Chem. C* **2017**, 121, 8033; *J. Am. Chem. Soc.* **2017**, 139, 14129). A sentence has been added to the Calculation Methods section to clarify the relativistic corrections adopted in this work as follow:

Paragraph 2, Page 24: “Scalar relativistic effect which includes the mass-velocity and Darwin correction terms, as implemented in VASP⁵⁸, was taken into account.”

Comment:

4. On page 18, it is stated that high molecule-to-Ln ratios (3:2 or higher) were frequently found in previous mononuclear coordination structures on surfaces. In this context, it is interesting to note that a 1:1 molecule-to-Ln ratio was achieved by reducing the number of molecules (by a factor of 5) through an on-surface oligomerization of dinitriles, resulting in a lanthanide superphthalocyanine.

Author reply:

We have cited the work mentioned by reviewer as Ref. 24 and revised the sentence mentioned by

the review as follow:

Paragraph 2, Page 21: “As a comparison, relatively high molecule-to-Ln (or molecular coordination site-to-Ln in the case of chelating ligands being engaged) ratios (3:2 or higher) were frequently found for the previously reported Ln-engaged mononuclear coordination structures constructed on surfaces¹⁴⁻²⁴, in which the single Ln atoms afforded multi (≥ 3)-fold coordination with the molecular ligands.”

Comment:

5. *Temperatures for the thermal treatment are reported as approximate values ("about 350 K"). Since precise temperatures may be important for the reproducibility of the experiments, error margins should be reported.*

Author reply:

We thank the reviewer for the suggestion. Our experiments were carried out with a temperature control within ± 20 K. We have added the statement of the temperature control precision to the Methods section as follow:

Paragraph 3, Page 23: “The thermal treatment of the samples was carried out with a temperature control within ± 20 K.”

Comment:

6. *It should be mentioned whether the differential electron density map in Figure 4c and 5d-f shows a projection or a cross section.*

Author reply:

The differential electron density maps present the cross sections parallel to the substrate surface of the differential electron densities of the structures. The cut positions of the cross sections are marked by the black dashed lines in Figs. 5a and 6a-c. The description of the differential electron density maps has been revised accordingly as follows:

Paragraph 1, Page 18: “The cross section of the differential electron density depicts the electron accumulation (red) and deficiency (blue) due to the formation of the coordination motif (see Methods for details of the calculation of differential electron density).”

Figure 5 caption, Page 16: “... (c) cross section of the differential electron density along the black dashed line in (a) of the Ce₂Au₂(CN)₆ motif on Au(111). ...”

Figure 6 caption, Page 21: “... Cross sections of the differential electron densities of (d) the Ce₂Au₂Py₄, (e) the Ce₃Au₄Py₆ and (f) the Ce₄Au₅(CN)₈ motifs along the black dashed lines in (a), (b)

and (c), respectively.”

Reviewer #3

Comment:

The paper describes a series of heterometallic clusters containing 2, 3 and 4 Cerium atoms bridged by Au adatoms prepared under UHV. The authors describe the formation of coordination structures based on Ce-Au clusters and pyridine or nitrile linkers on Au(111) and Au(100). Interestingly, Au adatoms serve as atomic “glue” to bridge the Ce atoms, as demonstrated by the DFT calculations, which play a crucial role in stabilizing the heterometallic clusters.

*The structures are analyzed by high-resolution STM, complemented by DFT calculations. The quality of the data is very good, though before I recommend publication some points have to be considered by the authors, as similar works have been recently reported (see for instance *Angew. Chem. Int. Ed.* 57, 4617-4621 (2018)) where Bi₃Cu₄ and Bi₇Cu₁₂ nanoclusters using pyridil linkers in 2D metal–organic networks have been observed.*

*- My main concern is related to the lack of information regarding the electronic properties of these systems (or at least of some of them), which I think are very interesting, as Ln-TM clusters involving multiple Ln atoms as coordination centers are scarcely reported, and needed to publish this work in a high-level impact factor journal such as *Nature Communications*.*

Author reply:

We have taken the three- and four-lobe structures constructed by the nitrile ligand **2** (structure shown in Fig. R3) and Ce trimers and tetramers as examples to explore their magnetic and electronic properties by STS experiments following the reviewer’s suggestion. However, we failed to get any magnetic signals of the Ce-containing clusters by high-resolution dI/dV measurement. It is similar to the cases of some surface-confined double-decker lanthanide phthalocyanine molecules in which the detection of the 4f electron states is elusive by STM measurement (*Nat. Commun.* **2011**, 2, 217; *Nat. Commun.* **2012**, 3, 953; *Phys. Chem. Chem. Phys.* **2015**, 17, 27019; *ACS Nano* **2018**, 12, 2991). The reason may lie in the inner-core nature of the 4f orbitals which leads to the little contribution of the 4f electron of Ce to the tunneling current. We neither found any Ce-specific dI/dV features in the large range (-4 to 3 V) dI/dV spectra. It may be explained by the strong hybridization between the Au substrate and the Ce-Au clusters.

Fig. R3. Chemical structure of ligand **2**.

Although the characterization of the magnetic and electronic properties of the Ce-Au clusters by STS was not accessible here, we believe that the on-surface coordination approach developed in this work should be applicable for the preparation of multinuclear heterometallic clusters involving other Ln metals, given the similarity in chemical properties of the Ln elements. In this context, for the Ln metals whose 4f electron states are reported to be detectable by STM techniques, such as Gd (*Science* **1997**, 275, 1767) and Ho (*Nature* **2017**, 543, 226), the investigation of their electronic and magnetic properties in the heterometallic clusters with varied nuclearities can be expected. We thank the reviewer for the inspiring advice for our study in the future.

Comment:

- I am aware of the difficulties of experimentally proving the presence of Au adatoms as "glue" between Ce atoms. Is there any difference between the simulated STM image shown in Figure 4b (where Au adatoms are having into account in the calculation) and the simulated STM image where those Au are not present? If yes, the authors should comment about it.

Author reply:

We appreciate the helpful suggestion by the reviewer. We have compared the simulated STM images of the three candidate models proposed as the coordination building block of the three-lobe structure, namely, $\text{Ce}_2(\text{CN})_6$ (Supplementary Fig. 11c), $\text{Ce}_2\text{Au}(\text{CN})_6$ (Supplementary Fig. 13b) and $\text{Ce}_2\text{Au}_2(\text{CN})_6$ (Fig. 5b). As a result, the STM simulation of $\text{Ce}_2\text{Au}_2(\text{CN})_6$ shows the best agreement with the experimental data. In contrast, the simulated STM image of $\text{Ce}_2\text{Au}(\text{CN})_6$ presents an asymmetric feature of the metallic center, while that of $\text{Ce}_2(\text{CN})_6$ shows one elliptical protrusion at the center rather than two separated round ones, both of which are inconsistent with the experimental observations. The comparison of the STM simulations supports the identification of the $\text{Ce}_2\text{Au}_2(\text{CN})_6$ motif as the coordination building block of the three-lobe structure. The STM simulations and discussion have been added to the main text and Supplementary Information as follows:

Paragraph 1, Page 16: “Moreover, the simulated STM image of the optimized $\text{Ce}_2\text{Au}_2(\text{CN})_6$ structure (Fig. 5b) was compared with those of $\text{Ce}_2(\text{CN})_6$ (Supplementary Fig. 11c) and $\text{Ce}_2\text{Au}(\text{CN})_6$ (Supplementary Fig. 13b). As a result, the STM simulation of $\text{Ce}_2\text{Au}_2(\text{CN})_6$ shows the best agreement with the experimental data, which supports the identification of the $\text{Ce}_2\text{Au}_2(\text{CN})_6$ structure as the

coordination motif to construct the three-lobe network.”

Supplementary Figure 11, Page S9, Supplementary Information:

Supplementary Figure 11. Theoretical results of the $\text{Ce}_2(\text{CN})_6$ structure. (a) Optimized model (upper: top view, lower: side view), (b) cross section of the differential electron density along the black dashed line in (a), and (c) simulated STM image of the $\text{Ce}_2(\text{CN})_6$ motif on Au(111).

Supplementary Figure 13, Page S11, Supplementary Information:

Supplementary Figure 13. Theoretical results of the $\text{Ce}_2\text{Au}(\text{CN})_6$ structure. (a) Optimized model (upper: top view, lower: side view) and (b) simulated STM image of the $\text{Ce}_2\text{Au}(\text{CN})_6$ motif on Au(111).

Figure 5, Page 16:

Fig. 5. Theoretical results of the $\text{Ce}_2\text{Au}_2(\text{CN})_6$ structure. (a) DFT optimized models (upper: top view, lower: side view), (b) simulated STM image and (c) cross section of the differential electron density

along the black dashed line in (a) of the $\text{Ce}_2\text{Au}_2(\text{CN})_6$ motif on Au(111). Simplified molecular ligands are employed.

Comment:

- *Experimental Ce-Ce distances in the clusters should be included and compared to the theoretical values shown in the manuscript.*

Author reply:

We have added the experimentally measured Ce-Ce distances of the coordinated clusters and compared them with the theoretical values as follows:

Paragraph 2, Page 15: “The Ce-Ce distance in the optimized model of the $\text{Ce}_2\text{Au}_2(\text{CN})_6$ motif (4.95 Å) is consistent with the experimental value (4.7 ± 0.4 Å) by taking the measurement error into account.”

Paragraph 2, Page 19: “The Ce-Ce distances in these models, i.e., 4.52 Å for $\text{Ce}_2\text{Au}_2\text{Py}_4$, 5.56 Å for $\text{Ce}_3\text{Au}_4\text{Py}_6$ and 4.77 Å for $\text{Ce}_4\text{Au}_5(\text{CN})_8$, are consistent with the experimentally measured values, that is, 4.5 ± 0.3 Å, 5.1 ± 0.5 Å and 4.9 ± 0.5 Å, respectively.”

In addition to the revisions mentioned above, we also made other changes to the main text and Supplementary Information. The phrases and sentences before and after the revision are listed below:

Authors and affiliations, Page 1

- Six authors have been added due to their contributions to the revision of the manuscript.
- Affiliations have been updated.

Figure 1 caption, Page 5:

- “... **1** [1,4-bis(4-pyridyl)-benzene], **2** [4,4’-dicyano-1,1’:3’,1’’-terphenyl], **3** [1,4-bis(4-pyridyl)-biphenyl] and **4** [1,3-bis(4-pyridyl)-benzene] ...” has been added.

Section heading, Page 5:

- *Before* RESULTS AND DISCUSSION
After RESULTS

Paragraph 2, Page 9:

- *Before* To further extend the range of conditions applicable for the on-surface preparation of the Ce-Au heterometallic clusters, ...
After To extend the range of conditions applicable for the on-surface preparation of the **multi-**

Ce-containing clusters, ...

Paragraph 1, Page 10:

- “With $\text{Ce}_2\text{Au}_2\text{Py}_4$ and $\text{Ce}_3\text{Au}_4\text{Py}_6$ as the building blocks” has been deleted.
- *Before* Figure 6 shows the three ordered structures assembled by the $\text{Ce}_2\text{Au}_2\text{Py}_4$ motifs comprising **3** or **4**, as illustrated by the superimposed molecular models.
After **Figure 4** shows the three ordered structures assembled by **the coordination between the Ce dimers and ligand 3** or **4**.
- *Before* the Ce_2Au_2 clusters
After the **two-Ce-containing** clusters
- *Before* Besides these $\text{Ce}_2\text{Au}_2\text{Py}_4$ -based structures, a low yield of the $\text{Ce}_3\text{Au}_4\text{Py}_6$ motifs formed by **3** was also observed (Supplementary Fig. 7d).
After Besides these **Ce-dimer**-based structures, a low yield of the coordination motifs formed by **the three-Ce-containing clusters and ligand 3** was also observed (Supplementary Fig. 6d).

Paragraph 1, Page 11:

- “Constructed by the $\text{Ce}_2\text{Au}_2\text{Py}_4$ motifs” has been deleted.
- “Constructed by the $\text{Ce}_3\text{Au}_4\text{Py}_6$ motifs” has been deleted.
- *Before* ... formed by **1** and the Ce-Au clusters, ...
After ... formed by **the pyridine ligand 1** and the **two- or three-Ce-containing** clusters, ...
- “Constructed by the $\text{Ce}_2\text{Au}_2(\text{CN})_6$ and $\text{Ce}_4\text{Au}_5(\text{CN})_8$ motifs, respectively,” has been deleted.
- *Before* ... formed by **2** and the Ce-Au clusters were all obtained on Au(100).
After ... formed by **the nitrile ligand 2** and the **two- or four-Ce-containing** clusters were all obtained on Au(100).

Subheading, Page 11:

- *Before* DFT Investigations of the Coordinated Ce-Containing Clusters.
After **Structures of the Coordinated Multi-Ce-Containing Clusters.**

Paragraph 2, Page 11:

- *Before* To understand the unexpected formation of the multi-Ce-containing clusters, DFT calculations were carried out.
After To **elucidate the structure of the multi-Ce-containing clusters, as well as to figure out the intra-cluster and cluster-ligand interactions, combined experimental and theoretical studies** were carried out.

- “The results reveal the heterometallic nature of the clusters, showing that the Ce atoms in the clusters are stabilized by the bridging Au adatoms, as demonstrated below.” has been added.

Paragraph 2, Page 12:

- “Next, we take the molecular ligands into account to demonstrate that the Ce atoms coordinated with the molecules are bridged by Au adatoms to form the multinuclear clusters observed in this work. The experimental evidence comes from both the STM and XPS results.” has been added.

Paragraph 1, Page 17:

- *Before* ..., which is in accordance with the significant difference in electronegativity of Ce and Au atoms: Ce is highly electropositive while Au is the most electronegative metal on the Pauling scale³⁵.

After ..., which is similar to the situation in the Ce-Au binary systems as mentioned above.

Figure 6 caption, Page 20:

- “(Upper: top view, lower: side view)” has been added.

Section heading, Page 21:

- “DISCUSSION” has been added.

Paragraph 1, Page 21:

- “Firstly” has been added.
- *Before* In all, the achievement of the various ordered coordination assemblies of the Ce-Au clusters with different molecules indicates the feasibility of utilizing different molecular ligands to tune the organization and periodicity of the Ce-Au cluster arrays.

After The different coordination behaviors of the pyridine and nitrile ligands with the Ce-Au clusters give rise to the various ordered coordination assemblies observed in this work, showing the feasibility of utilizing different molecular ligands to tune the organization and periodicity of the Ce-Au cluster arrays.

Section heading, Page 23:

- The section heading “CONCLUSION” has been deleted

Subheading, Page 23:

- *Before* Sample Preparation and STM Measurements
- *After* Sample Preparation, STM and XPS Measurements

Paragraph 2, Page 23:

- “The samples were prepared in UHV preparation chambers connected with either the STM or XPS chamber.” has been added.
- “All” has been deleted

- “All XPS experiments were carried out in a SPECS system with a monochromatic Al K α X-ray source ($h\nu = 1486.6$ eV). The base pressure of the system is lower than 5×10^{-10} Torr. The Au 4f $_{7/2}$ peak of a clean Au(111) substrate was used to calibrate the spectrometer before experiments. The Au 4f $_{5/2}$ peak of the Au(111) substrate at 87.7 eV was employed as an internal standard to calibrate the binding energy scale for this work.” has been added.

Equations, Pages 24 and 25:

- The equations are numbered.

References, Pages 26-31:

- Refs. 3, 24, 33-38, 40-46 and 58 have been added and the references are reordered.

Acknowledgments, Page 31:

- Funding numbers have been updated.

Author contributions, Page 31:

- Author contributions have been updated.

Page S1, Supplementary Information:

- Author list and affiliations have been added.

Section heading, Page S2, Supplementary Information:

- “**SUPPLEMENTARY FIGURES**” has been added.

Supplementary Figure 6 caption, Page S5, Supplementary Information:

- *Before* the Ce₃Au₄Py₆ motifs
After the coordination motifs comprising the three-Ce-containing clusters

Supplementary references, Page S14, Supplementary Information:

- Two references have been added.

REVIEWERS' COMMENTS

Reviewer #1 (Remarks to the Author):

The authors adequately addressed all my concerns in a fully convincing manner. I am impressed by the additional efforts they invested to support their conclusions now much more convincingly.

Reviewer #2 (Remarks to the Author):

The authors have responded very constructively to the reviewers' comments and have considerably improved an already very good manuscript, elevating it to an excellent level. Most importantly, the authors performed new experiments and added new data (XPS, STM) and improved the calculations by employing stricter convergence criteria in DFT and by selecting more appropriate points of reference for considerations of the stability of the clusters. With these new data and the corresponding discussion, the major claims of this study are now very well substantiated. Publication of the impressive and highly original work in Nature Communications is recommended.

J. Michael Gottfried

Reviewer #3 (Remarks to the Author):

Comments to the Author

The authors have satisfactorily replied to my concerns, therefore I would recommend publication after a minor revision:

- Include in the main text the discussion about the electronic properties of the studied system, i.e. three- and four-lobe structures constructed by the nitrile ligand 2.

In the following, we'll address one by one the reviewer's comments. All the responses are typed in blue. All the changes in the main text and Supplementary Information are highlighted in red.

Reviewer #3

Comment:

The authors have satisfactorily replied to my concerns, therefore I would recommend publication after a minor revision:

- Include in the main text the discussion about the electronic properties of the studied system, i.e. three- and four-lobe structures constructed by the nitrile ligand 2.

Author reply:

We have added the discussion about the electronic properties of the Ce-Au clusters to the main text as follow:

Paragraph 2, Page 23: “Last but not least, we have taken the three- and four-lobe structures constructed by the $\text{Ce}_2\text{Au}_2(\text{CN})_6$ and $\text{Ce}_4\text{Au}_5(\text{CN})_8$ motifs, respectively, as examples to explore the magnetic and electronic properties of the Ce-Au clusters by STS experiments. However, we failed to get any magnetic signals of the Ce-Au clusters by high-resolution dI/dV measurement. The similar situations have been reported for some surface-confined double-decker lanthanide phthalocyanine molecules whose 4f electron states were elusive to be detected by STS measurement⁵⁹⁻⁶². The reason may lie in the inner-core nature of the 4f orbitals which leads to the little contribution of the 4f electron of Ce to the tunneling current. We neither found any Ce-specific dI/dV feature in the large range (-4 to 3 V) dI/dV spectra. It may be explained by the strong hybridization between the Au substrate and the Ce-Au clusters.”